

**Water uptake patterns of pea and barley responded to drought but not to cropping systems**
**Author list**: Qing Sun[1], Valentin H. Klaus[1], Raphaël Wittwer[2], Yujie Liu[1], Marcel G.A. van der
Heijden[2,3], Anna K. Gilgen[1], Nina Buchmann[1]
[1] Institute of Agricultural Sciences, ETH Zurich, 8092 Zurich, Switzerland
[2] Department of Agroecology and Environment, Agroscope, Zurich, Switzerland
[3] Department of Plant and Microbial Biology, University of Zurich, Zurich, Switzerland
**Author for correspondence**: Qing Sun
Tel: +41 44 632 34 13
Email: qing.sun@usys.ethz.ch; s.qing@outlook.com

**Highlights**

• Pea and barley shifted to shallower water uptake depths in response to drought.
• No niche differentiation found between pea and barley in a mixture under drought.
• No differences on changes in uptake depths by drought found among cropping systems.
• Thus, cropping systems did not compensate drought effects on water uptake patterns.





**Abstract**
Agricultural production is under threat of water scarcity due to increasingly frequent and severe
drought events under climate change. Whether a change in cropping systems can be used as an
effective adaptation strategy against drought is still unclear. We investigated how plant water
uptake patterns of a field-grown pea-barley (*Pisum sativum* L. and *Hordeum vulgare* L.) mixture, an
important fodder crop, responded to experimental drought under four cropping systems, i.e., organic
intensive tillage, conventional intensive tillage, conventional no-tillage, and organic reduced tillage.
Drought was simulated after crop establishment using rain shelters. Proportional contributions to
plant water uptake from different soil layers were estimated based on stable water isotopes using
Bayesian mixing models. Pea plants always took up proportionally more water from shallower
depths than barley plants. Water uptake patterns of neither species were affected by cropping
systems. Both species showed similar responses to the drought simulation and increased their
proportional contributions from shallow soil layer (0-20 cm) in all cropping systems. Our results
highlight the impact of drought on plant water uptake patterns for two important crop species and
suggest that cropping systems might not be as successful as adaptation strategies against drought as
previously thought.
**Keywords**: climate change, conservation tillage, FAST, organic farming, stable water isotope,
water uptake depth



## 1 Introduction

Due to climate change, drought events may occur more frequently and become more severe, and hence water scarcity is worsening in many regions of the world (Schewe *et al.*, 2014; IPCC, 2019). Thus, agriculture is facing increasing pressure to ensure food security under aggravating conditions (FAO, 2018; FAO, 2019). Although crop breeding has large potential to enhance agricultural productivity, it should certainly not be seen as the only option. Adapted crop management is discussed as an additional solution to mitigate yield loss under drought, either by sustaining plant growth or by enhancing soil water availability (Cochard, 2002; Bot & Benites, 2005; Kundel *et al.*, 2020). Therefore, there is a growing interest in organic farming and conservation tillage (i.e., no tillage or reduced tillage), as these management practices have been shown to be beneficial to soil health and water holding capacity, ecosystem stability, as well as environmental sustainability (e.g., Seitz *et al.*, 2019; Teasdale *et al.*, 2007; Hobbs *et al.*, 2008). However, an evaluation of different cropping systems as a means to support arable crops under drought is still urgently needed (IPCC, 2019).

Understanding plant water relations under drought plays an increasingly important role in promoting sustainable agriculture to secure food production (Penna *et al.*, 2020). Plant water uptake and water use, particularly during critical growing stages, greatly determine physiological processes, survival, and ultimately crop productivity (Boyer & Rao, 1984; Wang *et al.*, 2015). Although many studies reported plant water uptake patterns in response to drought over a broad range of species and environments (e.g., Prechsl *et al.*, 2015; Grossiord *et al.*, 2019; Rasmussen *et al.*, 2020; Ding *et al.*, 2021), only very few focused on arable agriculture (e.g., Zegada-Lizarazu *et al.*, 2006; Borrell *et al.*, 2014; Wu *et al.*, 2018) and none compared arable cropping systems. Moreover, these studies found contrasting responses of crop species to changing environments, illustrating the current gap of knowledge for cropping systems.

Plant water uptake mainly depends on soil water availability, root properties and distributions, as well as soil-plant interactions (von Freyberg *et al.*, 2020). Soil water availability depends on soil



physical characteristics and local climatic conditions. Root systems, including root distribution and
functionality, are affected by soil physical and nutritional conditions as well as plant growth stages
and species genetics. Soil-root interactions include hydrotropism, root damage caused by drying
soil, and soil water redistribution (Caldwell *et al.*, 1998; Whitmore & Whalley, 2009; Dietrich *et*
*al.*, 2017). Furthermore, plant water uptake patterns are highly dynamic and difficult to track. Since
the 1960s, stable water isotopes, i.e. oxygen and hydrogen isotopes, have been used in
ecohydrology studies (Gonfiantini *et al.*, 1965; Zimmermann *et al.*, 1967), e.g. to assess root water
uptake patterns (Rothfuss & Javaux, 2017), to detect foliar water uptake (Berry *et al.*, 2019), as well
as to partition evapotranspiration fluxes (Wang *et al.*, 2010). Stable water isotopes have since
become a helpful tool to identify plant water uptake sources and quantify source contributions
(Dawson & Ehleringer, 1991; Penna *et al.*, 2018). However, studies in agroecosystems have often
focussed on grassland species (e.g. Bachmann *et al.*, 2015; Prechsl *et al.*, 2015), much less on crop
species as reviewed by Penna *et al.* (2020).
Hence, our experimental field study investigated how different cropping systems, namely organic
vs. conventional farming with intensive vs. conservation tillage, affect plant water uptake patterns
under drought using stable water isotopes. We focused on a pea-barley (*Pisum sativum* L. and
*Hordeum vulgare* L.) mixture, an increasingly popular intercrop for fodder production (Gilliland &
Johnston, 1992). We aimed at understanding (1) if pea and barley grown in mixture differ in their
water uptake patterns, (2) how drought affects plant water uptake depths, and (3) if cropping
systems affect water uptake depths differently.
**2    Materials and Methods**
**2.1    Research site and experimental setup**
The research site is in Rümlang near Zurich (47.26° N, 8.31° E), and belongs to the Swiss federal
agricultural research station Agroscope. Long-term average annual precipitation at the site is 994
mm, and mean annual air temperature is 9.7 °C (1988 to 2017; MeteoSwiss, 2020). The soil at the
research site is a calcareous Cambisol with 23% clay, 34% silt, and 43% sand, and total soil carbon



content of 1.6 to 1.8% (Loaiza Puerta *et al.*, 2018). The plant available soil depth is 50-70 cm, and
no groundwater is accessible for plants (Kanton Zürich, 2020). Our study used a sub-set of plots in
the Farming Systems and Tillage Experiment which began in 2009 with a six-year crop rotation that
is typical for Swiss cropping systems (for details see Wittwer *et al.*, 2017). It combines
conventional (C) and organic (O) farming with intensive or soil conservation tillage practices. The
conventional systems are managed according to the "Proof of Ecological Performance" (PEP)
guidelines of the Swiss Federal Office for Agriculture (Swiss Federal Council, 2021), which allows
synthetic fertiliser and pesticide applications. The organic systems were managed following the
BioSuisse guidelines, prohibiting the use of mineral fertilisers and synthetic plant protection
products. Intensive tillage (IT) with a mouldboard plough to 20 cm depth followed by seedbed
preparation with a rotary harrow to 5 cm depth was applied in both conventional (C-IT) and organic
systems (O-IT). For conservation tillage, direct sowing and no soil management were implemented
in the no tillage conventional plots (C-NT) but glyphosate was sprayed before sowing of the main
crops for weed control. A disc or rotary harrow, which superficially disturbed the soil for weed
control, was used for reduced tillage in organically managed plots (O-RT) to a maximum depth of
10 cm. These four cropping systems were repeated in four blocks following a Latin square design.
Cropping system plots had an area of 6 m × 30 m.
In 2018, the same pea (*Pisum sativum* L. cv. 'Alvesta') and barley (*Hordeum vulgare* L. cv.
'Eunova') mixture was sown in all plots on 26 March and harvested on 12 July. No fertilisation was
applied in any of the treatments because the pea plants were expected to fix dinitrogen from the
atmosphere. Portable, tunnel-shaped rain shelters (metal frames of 3 m × 5 m base area and 2.1 m
height at the highest point) with transparent and ultraviolet light-transmissible plastic foil
(Gewächshausfolie UV5, 200 μm, Folitec Agrarfolien-Vertrieb, Germany) were installed to
simulate a drought period from 22 May to 28 June 2018. Shelters were open at both ends as well as
at both sides and had an opening at the top along the full length. This allowed extensive ventilation
and prevented temperature build-up (for technical details see Hofer *et al.*, 2016). Rain running



down the foil was collected in PVC half pipes and directed away from the plots (about 2 m). During
the drought treatment period, 34% of precipitation during the growing season in 2018 (from sowing
to harvest) was excluded from the drought subplots (Table 1). These drought subplots were
established in each cropping system and located directly next to control subplots which received
natural precipitation inputs, resulting in a split-plot layout. A total of 16 experimental plots (four
cropping systems × four replicates) with 32 subplots (16 plots × two water availability treatments)
were used in this study.
**2.2    Climatic data and soil water contents**
Precipitation and air temperature data (Table 1; Fig. 1) were obtained from a nearby weather
station, Zürich/Kloten (KLO, 47.48° N, 8.54° E, 4.6 km north of the research site, MeteoSwiss,
2020). Soil water content (SWC) was continuously measured and recorded at 10 and 40 cm depths
with two replicates per cropping system (EC-5, Decagon Devices Inc., Pullman, WA, USA; factory-
calibrated). Data were averaged at 10 min intervals by data loggers (CR1000 and CR216, Campbell
Scientific Ltd., Loughborough, UK), then averaged for daily values.
**2.3    Plant and soil water samples for stable isotope analysis**
Plant and soil samples were collected on 7 May, 25 June, and 11 July 2018, i.e., before the drought
treatment (BT), at the end of the treatment (ET), and after the treatment (AT), respectively. Pea was
not sampled AT due to progressed senescence. Root crowns were collected for stable isotope
analysis of plant xylem water as this part best reflects the mixture of water sources taken up from
the soil in herbaceous plants (Barnard *et al.*, 2006; von Freyberg *et al.*, 2020). Four to six
individuals were collected and pooled into one sample per species and subplot. Root crowns were
cleaned quickly to remove remaining soil and then immediately sealed in air-tight glass tubes (12-
ml exetainer, Labco Ltd., Ceredigion, UK). In parallel to the plant sampling, soil samples were
collected close to the sampled plants with a soil auger (1 cm diameter). The soil cores were
separated into six depth layers – 0-5, 5-10, 10-20, 20-30, 30-40, and 40-60 cm – and then
immediately sealed in glass tubes (18 ml, Schott AG, Mitterteich, Germany). All plant and soil



samples for stable water isotope analysis were kept in a cool box in the field and then stored
at -18 °C before extraction with cryogenic vacuum distillation (Ehleringer & Osmond, 1989).
**2.4    Stable water isotope analyses**
The oxygen and hydrogen stable isotope ratios ($\delta^{18}$O and $\delta^2$H) of extracted water samples were
analysed with an isotope ratio mass spectrometer (IRMS, DeltaplusXP, Finnigan MAT, Bremen,
Germany) using the methods described by Werner *et al.* (1999). All $\delta^{18}$O and $\delta^2$H values are
expressed relative to the Vienna Standard Mean Ocean Water (VSMOW-SLAP, Craig & Gordon,
1965; Gat, 2010) in parts-per-thousand (or "per mil", ‰; eq. 1):

$$\delta^{18}\text{O or }\delta^2\text{H} = \frac{R_{\text{SAMPLE}}}{R_{\text{STANDARD}}} - 1 \qquad (1)$$

where $R$ is the isotope ratio of the rare isotope to the abundant isotope ($^{18}$O/$^{16}$O or $^2$H/$^1$H). The long-
term precision of the quality-control standard *IsoLab 1* over the last four years was 0.22‰ for d$^{18}$O
and 0.59‰ for d$^2$H.
The isotopic composition of precipitation at the global scale shows a linear relationship between the
$\delta^{18}$O and $\delta^2$H of meteoric waters (Global Meteoric Water Line, GMWL; Craig, 1961), described by
the regression line in a "dual-isotope" $\delta^{18}$O-$\delta^2$H plot (eq. 2):

$$\text{GMWL: } \delta^2\text{H} = 8.2 \times \delta^{18}\text{O} + 11.7 \qquad (2)$$

Similarly, the Local Meteoric Water Line (LMWL) describes the isotopic composition in rainfall
for a specific location (Dansgaard, 1964). We fitted the long-term LMWL (1994 to 2017) with
monthly mean data from the closest GNIP station (Global Network of Isotopes in Precipitation,
Buchs Suhr, 47.37° N, 8.08° E, 34 km from the research site; IAEA, 2020; eq. 3), while the LMWL
of 2018 was fitted with data of precipitation samples collected at the research site (after Prechsl *et*
*al.*, 2014; eq. 4) during the growing season and data of 2018 from GNIP Buchs (Fig. S1):

$$\text{long-term LMWL: } \delta^2\text{H} = 7.9 \times \delta^{18}\text{O} + 6.4 \qquad (3)$$



$$\text{2018 LMWL: } \delta^2H = 8.3 \times \delta^{18}O + 12.7 \tag{4}$$

**2.5    Bayesian mixing model for plant water uptake**
Proportional contributions of soil water to plant water uptake (PC) from different depths were
estimated using mixing models from the R package 'simmr' (Parnell, 2020) within a Bayesian
framework based on code by Parnell *et al.* (2013). The $\delta^{18}O$ or $\delta^2H$ signatures of soil water from the
six soil layers were used as sources, and plant xylem water was considered the mixture for
modelling in each subplot at different sampling times, i.e., BT, ET, and AT. Missing replicates of
soil samples due to sampling difficulties (n = 5 in total) were filled with mean values of the other
replicates from the same cropping system and treatment to have balanced model inputs. The model
outputs consisted of 10 000 possible combinations of PC from different soil depths from four
Markov chain Monte Carlo Bayesian models with at least 300 000 iterations, 50 000 burns, and 100
times of thinning for each chain. The median of the model outputs on PC (MPC) from each soil
depth was calculated for each subplot and used for statistical analysis on plant water uptake depths.
Compared to the most frequent value of the model outputs, MPCs of all the sources usually sum up
closer to 1. To increase clarity of presentation, PC was grouped into three layers, namely shallow
(0-20 cm), middle (20-40 cm), and deep (40-60 cm) soil layers for further analyses. The PC values
from shallow and middle layers are the sum of PC from soil depths of 0-5, 5-10, and 10-20 cm, and
the sum of PC from soil depths of 20-30 and 30-40 cm, respectively. As $\delta^{18}O$ and $\delta^2H$ yielded
similar results, only the model outputs of $\delta^{18}O$ are described in detail in this paper.
**2.6    Data analyses**
For data analyses, the whole growing season was divided into three periods based on the drought
treatment, namely before the drought treatment (BT; 26 March to 21 May), the drought treatment
period itself (22 May to 28 June) which was sampled directly before the removal of shelters on 28
June (termed ET, end of treatment), and after the drought treatment (AT, 29 June to 12 July). All
statistical analyses were carried out using R (v3.6.2; R Core Team, 2020). The effects of cropping
systems, drought treatment, and species were tested with linear mixed models using the function



*lmer()* from the R package 'lmerTest' (Kuznetsova *et al.*, 2017). 'Cropping systems (CS)', 'drought
treatment (D)', and 'blocks' were three fixed factors (Dixon, 2016), interactive effects between
'CS' and 'D' with 'plots' (accounting for the split-plot design) were considered as random factors.
For variables measured on both pea and barley (i.e., stable isotopes of xylem water and MPC for BT
and ET), 'plant species', 'CS', 'D', and 'blocks' were tested as fixed factors considering interactive
effects among 'plant species', 'CS', and 'D' with 'plots' and 'subplots' as random factors.
Diagnostic plots were checked for normality and homoscedasticity of residuals for model
assumptions. Differences among cropping systems and between treatments or species were tested
by the Tukey HSD (honestly significant difference) test using the function *glht()*, from the R
package 'multcomp' (Hothorn *et al.*, 2008).

## 3  Results

### 3.1  Environmental conditions in drought and control subplots

Air temperatures in 2018 were very high compared to the long-term mean, in particular in May and
June, with a daily average air temperature of 15.8 and 18.8 ºC, respectively, while the long-term
(1988 to 2017) mean air temperatures in these two months were 13.9 and 17.2 ºC, respectively
(Table 1; Fig. 1). Annual precipitation was relatively low (Table 1), with no precipitation between
14 June and 2 July 2018, and an even more pronounced drought period in July (Fig. 1). Thus,
average daily soil water contents (SWC) in the control subplots ranged from 16% to 29% at 10 cm
depth and slightly higher, from 22% to 29%, at 40 cm depth, prior to the rain event on 3 July 2018.
After this rain event, SWC increased in all cropping systems at both depths (Fig. 2a, b). Variations
in SWC among cropping systems were small, particularly during the natural drought in June.
SWC in drought subplots of all cropping systems decreased continuously during the drought
treatment (22 May to 28 June 2018), averaging to 13% at 10 cm and to 19% at 40 cm soil depth
(Fig. 2 c, d). SWC at 10 cm did not show any pronounced differences among cropping systems,
while SWC at 40 cm tended to be slightly higher in cropping systems with conservation tillage (O-
RT and C-NT) compared to systems with intensive tillage (O-IT and C-IT; Fig. 2b, d).



### 3.2    Stable isotopes in soil water and plant xylem water


In the dual-isotope space, stable oxygen and hydrogen isotope ratios of soil and plant xylem waters
were strongly related with each other ($R^2$ = 0.89 and 0.85, respectively; Fig. S1) and generally fell
below the local meteoric water line (LMWL) of 2018, representing evaporation. Stable isotope
signatures of xylem water were lower than the LMWL but higher than those of soil water,
indicating that xylem water isotope signatures were mixtures of the original source precipitation and
the pool of soil water, affected by different degrees of fractionation.
The stable water isotope profiles of soil water showed a characteristic pattern at all times, for all
cropping systems and both treatments, with most enriched values in the uppermost soil and
increasingly depleted values with increasing soil depth (Table S1; Fig. 3 for $\delta^{18}O$; Fig. S2 for $\delta^2H$).
The drought treatment showed no significant effects before the treatment (BT) for $\delta^{18}O$ nor $\delta^2H$
(except for $\delta^2H$ at 20-30 cm; Table 2). In contrast, at the end of the drought treatment (ET), soil
water $\delta^{18}O$ values from 20-60 cm (20-30, 30-40, and 40-60 cm) as well as $\delta^2H$ values from all
depths were strongly affected by the drought treatment (all $P < 0.05$; Table 2), with more depleted
signatures in the drought than in control subplots due to the exclusion of more enriched summer
precipitation. Even after the shelters were removed and the treatment had been finished (AT), the
drought treatment still significantly affected both $\delta^{18}O$ and $\delta^2H$ of soil water, albeit only in deeper
soil depths (30-40 and 40-60 cm for $\delta^{18}O$ and 40-60 cm for $\delta^2H$; all $P < 0.05$; Table 2). Overall,
cropping systems did not significantly affect the stable isotopic signatures in soil water at any time
(Table 2).
Pea xylem water was always significantly more enriched in $^{18}O$ and $^2H$ compared to barley (all $P <$
0.001; Table S2). The $\delta^{18}O$ values in xylem water for pea ranged between -8.8‰ and -5.7‰, and
significantly lower between -10.1‰ and -5.8‰ for barley (averages per cropping system, treatment,
and time; Table 3; Table S2). Similarly, the $\delta^2H$ values in xylem water for pea ranged
between -65.6‰ and -52.1‰, and significantly lower between -74‰ and -47.1‰ for barley (Table
3; Table S2). Overall, isotopic signatures in xylem water became more enriched in $^{18}O$ and $^2H$



during the growing season for both pea and barley (Fig. 3, Table S2, Fig. S2). On average, the
xylem $\delta^{18}$O for pea was -8.5‰ before the treatment (BT) and -7.2‰ at the end of the treatment
(ET), compared to -9.8‰ (BT), -8.8‰ (ET), and -6.3‰ after the treatment (AT) for barley. While
average $\delta^{2}$H values for pea were -64.1‰ (BT) and -57.6‰ (ET), $\delta^{2}$H values averaged -72.2‰
(BT), -68.6‰ (ET), and -50.8‰ (AT) for barley (Fig. 3; Table S1; Fig. S2). Since there was a
strong relationship between $\delta^{18}$O and $\delta^{2}$H in xylem water (Fig. S1; $R^2$ = 0.85), our analyses are
mainly focused on $\delta^{18}$O in the text (but see Table 3, Table S2, and Fig. S2 for analyses on $\delta^{2}$H).
For pea, cropping systems did not significantly affect $\delta^{18}$O nor $\delta^{2}$H in xylem water at either time
(BT and ET; Table S2), while the drought treatment significantly affected the isotopic signatures of
$^{18}$O only at the end of treatment (ET: $P$ = 0.022; no interactions between cropping systems and
drought treatment: $P$ = 0.085; Table S2). $^{18}$O in pea xylem water were significantly more enriched
in the drought than in the control subplots (on average, $\delta^{18}$O of -6.9‰ and -7.7‰, respectively).
In contrast to pea, cropping systems significantly affected $\delta^{18}$O in barley xylem water (ET: $P$ =
0.035; Table S2). The drought treatment significantly affected the isotope signatures of both $^{18}$O
and $^{2}$H at the end of treatment (ET: both $P$ < 0.01; no interactions between cropping systems and
drought treatment; Table S2). However, unlike pea, the xylem water of barley showed significantly
lower $\delta^{18}$O values in drought than in control subplots for all cropping systems (on average, -9.0‰
and -8.6‰, respectively), although the difference was small (Table S2). A similar pattern was also
observed for $\delta^{2}$H at the end of treatment (ET), with significantly lower values on average in drought
than in control subplots (ET: -71.8‰ and -65.4 ‰, respectively).
**3.3    Modelled plant water uptake depths**
The outputs of the Bayesian mixing model on the proportional contribution to total plant water
uptake (PC) showed highly significantly different behaviours of pea and barley, mirroring some of
the differences seen in the xylem water isotopic signatures of these two species (Fig. 4; Fig. 5).
Since frequency density distributions provide not only one estimate per soil depth, but a full
frequency distribution, the medians were calculated for each soil depth to assist in the analyses



(Table S3 for results from $\delta^{18}$O; Table S4 for results from $\delta^2$H). As both stable isotope signatures
showed similar results, we here focus on results derived from $\delta^{18}$O only. In addition, we grouped
the uptake depths into shallow (0-20 cm as sum of 0-5, 5-10, and 10-20 cm), middle (20-40 cm as
sum of 20-30 and 30-40 cm), and deep (the original 40-60 cm) soil layers (Table 4; Table 5).
Overall, both species took up water from the entire soil profile studied (0 to 60 cm soil depth), albeit
with different proportions depending on species, time (i.e., BT, ET, and AT) and treatment (i.e.,
control vs. drought; Table 4; Table 5).
For pea, soil water contributions to total plant water uptake decreased with increasing soil depth in
both control and drought subplots before (BT) and at the end of the treatment (ET) for all cropping
systems (Fig. 4). The median of PC values (MPC) differed significantly among shallow (0-20 cm),
middle (20-40 cm), and deep (40-60 cm) layers, averaging 47%, 33%, and 16%, respectively, for
both treatments and all cropping systems (BT; Table 5; Fig. 4a, c). At ET, pea plants subjected to
drought significantly shifted their water uptake to even higher contributions from the shallow layer
(67%) and less uptake from middle (22%) and deep (8%) soil layers compared to BT (Table 5; Fig.
4d; Table S5). Pea plants in control subplots did not display such a significant shift, but remained
with average MPC from shallow, middle, and deep soil layers of 52%, 31%, and 14%, respectively
(Table 5; Fig. 4b; Table S5). Cropping systems did not significantly affect MPC before (BT) or at
the end of (ET) treatment (also no interactions between cropping systems and drought, Table 5; Fig.
4d).
In contrast to pea, barley plants showed very different water uptake patterns before the treatment
(BT), with significantly lower PC from the shallow soil layer compared to the middle and deep
layers. For barley, MPC values averaged 19%, 44%, and 35% for shallow, middle, and deep soil
layers, respectively, for both treatments and all cropping systems (Fig. 5a, d). However, at the end
of the treatment (ET), barley plants significantly increased the contributions from the shallow layer
in drought subplots, similar to pea (Table 5; Fig. 5e; Table S5), resulting in MPC values of 38%,
41%, and 18% from shallow, middle, and deep soil layers, respectively. The MPC further shifted



after the treatment (AT) to values of 62%, 27%, and 10% from shallow, middle, and deep layers,
respectively (Fig. 5f). Also in control subplots, barley plants showed the same significant shift from
BT to ET, with MPC values at ET of 35%, 34%, and 29% from shallow, middle, and deep layers,
respectively (Table 5; Fig. 5b; Table S5), and from ET to AT with MPC values AT of 59%, 29%,
and 12% from shallow, middle, and deep layers, respectively (Table 5; Fig. 5c; Table S5). Similar
to pea, barley water uptake patterns were not significantly affected by cropping systems (Table 5).
Overall, MPC values from shallow and deep layers for pea and barley were positively correlated (r
= 0.64 and 0.55, respectively; Fig. S3). This means when barley took up more water from the
shallow layer, so did pea.
Organic as well as reduced/no tillage cropping systems are discussed as adaptation strategies under
climate change conditions to ensure arable crop production. Thus, we analysed plant water uptake
depths in drought subplots at the end of treatment (ET) more in detail, although cropping systems
showed no significant effects on water uptake depths for either species and no interactions occurred
between cropping systems and drought treatment (Table 5). Pea plants in both intensive systems (C-
IT and O-IT) showed significantly higher (O-IT: 77%) or similar (C-IT: 65%) contributions to total
water uptake (as MPC) from the shallow layer (0-20 cm) compared to conservation tillage systems
(64% in both C-NT and O-RT; Table 5; Fig. 4d). Conversely, contributions from the middle layer
(20-40 cm) for pea at the end of treatment (ET) were only 15% in O-IT compared to 24% in the
other three cropping systems (O-RT, C-IT, and C-NT). Differences among cropping systems under
drought were even smaller for barley than for pea (Table 5; Fig. 5e). MPC values of barley for
uptake from the shallow layer were 47% (C-IT), 39% (O-RT), 31% (O-IT), and 32% (C-NT).
Conversely, contributions from the middle layer were the largest in C-NT (47%), followed by O-IT
(44%) and O-RT (41%), and lowest in C-IT (34%).
**4     Discussion**
Root water uptake patterns are often discussed for their important role in plant water relations, but
only few studies considered arable crop species (Penna *et al.*, 2020). In addition, most studies on



responses of crop root water uptake patterns to drought took place in pots or under controlled
conditions (e.g., Zegada-Lizarazu & Iijima, 2004; Araki & Iijima, 2005), so that information on
field conditions is particularly scarce, except maize (Ma & Song, 2016), wheat (Ma & Song, 2018),
oilseed rape, and barley in monoculture (Wu *et al.*, 2016). Furthermore, studies comparing the role
of different cropping systems for crop water uptake are completely lacking. Here, we showed for
the first time that root water uptake patterns of field-grown pea and barley in mixture responded to
drought but not to different cropping systems. Subjected to a pronounced drought period (37 d
without precipitation), both crop species shifted to relying more on shallow soil layer (0-20 cm) for
water uptake. This drought response was independent of the cropping system, i.e. organic vs.
conventional farming or intensive vs. conservation tillage.
Previous research on root water uptake patterns in crop as well as grassland species showed
ambiguous responses to drought. For some species, root water uptake depth was dependent on root
distribution during wet periods, but on soil water availability during dry periods (Sprenger *et al.*,
2016). Therefore, utilising more water from deep than from shallow soil layer is typically the
anticipated drought response, such as barley in monoculture (Wu *et al.*, 2018), maize (Ma & Song,
2016), wheat, rice, soybean (Zegada-Lizarazu & Iijima, 2004), or chickpea (Purushothaman *et al.*,
2017). However, other studies reported that crop and grassland species do not take up water from
deeper depths under drought but even absorb more water from shallow soil layer (e.g., barley in
monoculture, maize, pigeon pea, cowpea; Zegada-Lizarazu & Iijima, 2004), or grassland species
(Hoekstra *et al.*, 2014; Prechsl *et al.*, 2015; Wu *et al.*, 2016). This is in accordance with our results
in which both pea and barley increased their proportional water uptake from shallow layer (0-20
cm) at the end of treatment (ET) in the drought subplots. Although soil water contents (SWC) were
still higher at 40 cm than at 10 cm at the end of the treatment (ET; Fig. 2c, d), SWC at 40 cm and 10
cm depths were both very low. Thus, the whole soil profile showed very low water availability at
the end of the treatment (ET), and fine root distributions most likely dominated plant water uptake
patterns.



Rooting profiles for legumes with increased proportions of deeper roots under drought, e.g., below
23-30 cm, have been reported (Benjamin & Nielsen, 2006; Purushothaman *et al.*, 2017), although
different responses in root growth to drought were found among different varieties (Kashiwagi *et*
*al.*, 2006; Kumar *et al.*, 2012; Purushothaman *et al.*, 2017). The architecture of legume root systems
is strongly affected by rhizobia, which typically find better living conditions in terms of oxygen and
nitrogen concentrations higher up in the soil profile than at greater depths (Concha & Doerner,
2020), also in dry soils. Moreover, barley grown under drought conditions has been reported to
develop proportionally more shallow roots (0-20 cm depth) relative to deeper soil depths (Carvalho
*et al.*, 2014). Also, studies on grassland plants (both legume and grass species) found increasing
root biomass production in shallow soil depths (0-15 cm) in response to drought (e.g., Prechsl *et al.*,
2015). Moreover, shifting to shallower water uptake depths during drought might actually be
beneficial for nutrient acquisition (Querejeta *et al.*, 2021), since not only concentrations of soil
water and atmospheric $N_2$ are higher in the top soil than in the deeper soil, but also litter inputs for
N mineralisation. Although we did not investigate root distributions for either crop species, they
most likely follow such evolutionary strategies as well, in addition to recent crop breeding efforts
leading to less deep root systems in general (Canadell *et al.*, 1996; Thorup-Kristensen *et al.*, 2020).
Thus, besides the low soil moisture within the entire soil profiles, root systems biology clearly
contributed to the shift towards shallower water uptake depths under drought for both pea and
barley in this study.
The year 2018 was characterised by low precipitation during our experimental period, which
affected pea and barley plants in our control subplots differently (Fig. 6a, b). While pea did not shift
its water uptake pattern (Fig. 6a; Table S5), barley grown in the control subplots reacted very
similar to the natural 11-d dry period (before the ET sampling, 14 to 25 June; Fig. 2) as barley
subjected to our drought treatment, namely with a clear shift from deep (40-60 cm) to shallow (0-20
cm) soil layer (Fig. 6b, d; Table S5). However, barley still relied more on water uptake from the
deep soil layer during this natural dry period than under the experimental drought ($P = 0.017$; Table



5). Hence, these different reactions of the two species to the dry period clearly indicated that barley
was more susceptible than pea even to a mild water stress. This observation is fully in line with
measurements of stem hydraulic traits (i.e., loss of xylem conductance) from the same experiment
(Sun et al., 2021). Barley plants lost xylem conductance much earlier than pea plants when xylem
water potentials decreased. In addition, legumes like pea can maintain low stomatal conductance to
avoid water stress without compromising photosynthesis when growing under conditions with
limiting water supply, due to their high foliar N concentrations (Adams et al., 2018). This adds to
the hydraulic trait benefits of pea and explains why pea was less affected by the natural dry period.
Nevertheless, as shown in our study, if severities and frequencies of droughts increase in the future,
one can expect negative consequences not only on the performance of barley, but also of pea
(Martin & Jamieson, 1996).
Moreover, the two species growing together in the pea-barley mixture showed distinct niches for
root water uptake before drought, with pea relying more on water from shallow (0-20 cm) and
barley from deep (40-60 cm) soil layers, in accordance with resource partitioning in the absence of
water limitation as observed in intercrops, e.g., pearl millet and cowpea (Zegada-Lizarazu et al.,
2006) and in mixed-species grasslands (e.g., Hoekstra et al., 2014). However, the niches became
more similar under drought conditions, contradicting ecological theory which postulates more
pronounced niche differentiation and less niche overlap under stressful conditions, such as during a
drought (see Nippert & Knapp, 2007; Silvertown et al., 2015; Guderle et al., 2018). However, our
results were in line with results from biodiversity studies in temperate grasslands (Bachmann et al.,
2015; Barry et al., 2020; Hoekstra et al., 2014) which also did not show niche differentiation in
response to increased competition or drought_ENREF_5. Thus, further detailed knowledge on the
dynamics of intercrop water uptake patterns is needed to solve this contradiction and to decrease the
uncertainty for arable crop production now and under future climate conditions.
As global agriculture has already been considerably compromised by and become increasingly
sensitive to climate change (Ortiz-Bobea et al., 2021), farming practices such as organic





management and conservation tillage are being discussed widely. They have been shown to
improve general soil conditions compared to conventional management and intensive tillage,
particularly under drought (Bot & Benites, 2005; Gomiero et al., 2011; Choudhary et al., 2016). For
instance, organic management and conservation tillage can increase soil water holding capacity,
therefore providing higher water availability than conventional management and intensive tillage
(e.g., Colombi et al., 2019; Kundel et al., 2020). In this study, the systems with conservation tillage
(C-NT and O-RT) indeed showed slightly higher SWC than systems with intensive tillage (C-IT
and O-IT) at 40 cm (Fig. 2d). However, this did not result in any benefit for root water uptake
patterns of pea and barley against drought. Water uptake of both species shifted to the shallow layer
(0-20 cm) in all cropping systems under drought, without cropping system effects or interactive
effects between cropping systems and drought treatment. The relatively short period that annual
crop species are growing under these conditions might limit the potential benefits from improved
soil conditions present in those systems (e.g., Dennert et al., 2018; Loaiza Puerta et al., 2018;
Schluter et al., 2018). Although it remains to be seen if the observed behaviour of a pea-barley
mixture also holds true for other crop species, our results clearly challenge the potential of cropping
management under temperate climate as a tool to adapt arable agriculture to climate change.
**5    Conclusions**
Water uptake patterns of pea and barley both shifted under drought in all cropping systems and both
species relied more on water from the shallow soil layer (0-20 cm) than on water from deeper in the
soil profile. This was also the case for organic and reduced/no tillage cropping systems, which are
often discussed as beneficial for crop performance, particular under water-limited conditions, and
are thus suggested as adapted cropping management practices under a future climate. However, in
this study, we showed for the first time that cropping systems could not counteract the drought
effects on plant water uptake patterns for pea and barley grown in mixture. It remains to be seen if
this observation also holds true for other, major crops grown under water-limited conditions.



**Funding**

This work was supported by the Mercator Research Program of the ETH Zurich World Food System Center and the ETH Zurich Foundation.

**Acknowledgement**

The authors would like to thank the RELOAD team, in particular Emily Oliveira and Ivo Beck for their great technical and logistical support, Reto Zihlmann from ETH Zurich statistical consulting service for advise on statistical analyses, Yangyang Jia, Gicele Silva Duarte Sa, Xingyu Hu, and Ming Yi for their assistance in fieldwork, and Elham Rouholahnejad for constructive scientific discussion. Roland A. Werner and Annika Ackermann are greatly acknowledged for measurements of many water isotope samples.

**Author Contribution**

NB, AKG, RW, and MH designed the study; QS analysed the data; QS, AKG, and NB wrote the first drafts of the manuscript; all authors discussed the results, revised, and agreed on the final version of the manuscript.

**Conflict of Interest**

None declared.

**Supporting Information**

Additional supporting information can be found in the online version of this article.

Table S1 Stable water isotope values ($\delta^{18}O$ and $\delta^2H$, ‰) of soil in control and drought subplots under different cropping systems.

Table S2 Effects of cropping systems, drought treatment and the interaction on stable isotope data ($\delta^{18}O$ and $\delta^2H$, ‰) of pea and barley as well as mean $\pm$ 1 SE for each species in control and drought subplots under different cropping systems.





**Data Availability Statement**
The data that support the findings of this study will be openly available in the ETH Zurich
Repository at https://www.research-collection.ethz.ch/.





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





Table 1 Precipitation and air temperature data from a nearby weather station, Zürich/Kloten (KLO,
47.48° N, 8.54° E, 4.6 km north of the research site, MeteoSwiss, 2020) as well as dates for the
growing season (from sowing to harvest) and treatment periods in 2018.

|  | Date | Total precipitation (mm) | Mean air temperature (℃) |
|---|---|---|---|
| Long-term annual (1988-2017) | 1 January to 31 December | 994 | 9.7 |
| Annual (2018) | 1 January to 31 December | 856 | 11.2 |
| Long-term May (1988-2017) | 1 to 31 May | 105 | 13.9 |
| May 2018 | 1 to 31 May | 102 | 15.8 |
| Long-term June (1988-2017) | 1 to 30 June | 102 | 17.2 |
| June 2018 | 1 to 30 June | 40 | 18.8 |
| Growing season 2018 | 26 March to 12 July | 231 | 15.7 |
| Before drought treatment | 26 March to 21 May | 108 | 12.7 |
| End of drought treatment | 22 May to 28 June | 79 (34% of the growing season) | 18.7 |
| After drought treatment | 29 June to 12 July | 44 | 20.0 |




Table 2 Effects of cropping systems (CS, df = 3), drought treatment (D, df = 1) and the interaction
(CS × D, df = 3) on stable water isotopes ($\delta^{18}O$ and $\delta^2H$) in different soil depths before the drought
treatment on 7 May, at the end of treatment on 25 June, and after the treatment on 11 July (in 2018
tested by linear mixed models.

| Isotope | Depth (cm) | CS | D | CS × D | Blocks |
|---|---|---|---|---|---|
| | | | Before drought treatment | | |
| $\delta^{18}O$ | 0-5 | 0.580 | 0.555 | 0.458 | 0.788 |
| | 5-10 | 0.119 | 0.276 | 0.073 | 0.367 |
| | 10-20 | 0.489 | 0.836 | 0.516 | 0.459 |
| | 20-30 | 0.201 | 0.164 | 0.128 | 0.069 |
| | 30-40 | 0.135 | 0.437 | 0.882 | 0.311 |
| | 40-60 | 0.960 | 0.898 | 0.845 | 0.404 |
| $\delta^2H$ | 0-5 | 0.831 | 0.120 | 0.423 | 0.982 |
| | 5-10 | 0.158 | 0.118 | 0.056 | 0.516 |
| | 10-20 | 0.467 | 0.416 | 0.574 | 0.571 |
| | 20-30 | 0.105 | **0.026** | 0.064 | 0.181 |
| | 30-40 | 0.089 | 0.125 | 0.959 | 0.308 |
| | 40-60 | 0.560 | 0.291 | 0.853 | 0.436 |
| | | | End of drought treatment | | |
| $\delta^{18}O$ | 0-5 | 0.316 | 0.835 | 0.253 | 0.367 |
| | 5-10 | 0.189 | 0.247 | 0.766 | 0.168 |
| | 10-20 | 0.080 | 0.603 | 0.920 | 0.673 |
| | 20-30 | 0.898 | **<0.001** | 0.852 | 0.94 |
| | 30-40 | 0.437 | **<0.001** | 0.651 | 0.954 |
| | 40-60 | 0.073 | **0.008** | 0.616 | 0.594 |
| $\delta^2H$ | 0-5 | 0.295 | **<0.001** | 0.168 | 0.479 |
| | 5-10 | 0.330 | **0.005** | 0.859 | 0.215 |
| | 10-20 | 0.091 | **0.029** | 0.700 | 0.659 |
| | 20-30 | 0.889 | **<0.001** | 0.863 | 0.820 |
| | 30-40 | 0.388 | **<0.001** | 0.551 | 0.970 |
| | 40-60 | 0.136 | **0.006** | 0.469 | 0.809 |
| | | | After drought treatment | | |
| $\delta^{18}O$ | 0-5 | 0.393 | 0.059 | 0.848 | 0.291 |
| | 5-10 | 0.730 | 0.672 | 0.111 | **0.031** |
| | 10-20 | 0.538 | 0.612 | 0.734 | 0.993 |
| | 20-30 | 0.933 | 0.136 | 0.936 | 0.944 |
| | 30-40 | 0.881 | **0.048** | 0.979 | 0.772 |
| | 40-60 | 0.751 | **0.001** | 0.560 | 0.380 |
| $\delta^2H$ | 0-5 | 0.776 | 0.056 | 0.667 | 0.421 |
| | 5-10 | 0.117 | 0.958 | 0.649 | 0.636 |
| | 10-20 | 0.228 | 0.887 | 0.926 | 0.815 |
| | 20-30 | 0.710 | 0.104 | 0.888 | 0.705 |
| | 30-40 | 0.877 | 0.050 | 0.919 | 0.699 |
| | 40-60 | 0.841 | **<0.001** | 0.493 | 0.484 |

CS and D were tested as two fixed effect factors for all subplots (*P* values are given). Significant
differences are shown in bold (*P* < 0.05).





Table 3 Effects of species (df = 1), cropping systems (CS, df = 3), drought treatment (D, df = 1) and
the interaction (species × CS, df = 3; species × D, df = 1; CS × D, df = 3; species x CS × D, df = 3)
on stable water isotopes ($\delta^2$H and $\delta^{18}$O) of pea and barley before the drought treatment on 7 May
and at the end of treatment on 25 June in 2018 tested by linear mixed models.

| Factor | Before drought treatment | | End of drought treatment | |
|---|---|---|---|---|
| | $\delta^{18}$O | $\delta^2$H | $\delta^{18}$O | $\delta^2$H |
| Species | **<0.001** | **<0.001** | **<0.001** | **<0.001** |
| CS | 0.251 | 0.382 | **0.038** | 0.055 |
| D | 0.106 | **<0.001** | 0.143 | **0.001** |
| Species × CS | 0.184 | **0.023** | 0.312 | 0.348 |
| Species × D | 0.796 | 0.486 | **0.004** | **0.016** |
| CS × D | 0.190 | 0.117 | 0.051 | 0.081 |
| Species × CS × D | 0.290 | **0.045** | 0.120 | 0.070 |
| Blocks | 0.485 | 0.599 | **0.004** | 0.162 |


CS and D were tested as two fixed effect factors for all subplots (*P* values are given). Significant
differences are shown in bold (*P* < 0.05).





Table 4 Effects of species (df = 1), cropping systems (CS, df = 3), drought treatment (D, df = 1) and
the interaction (species × CS, df = 3; species × D, df = 1; CS × D, df = 3; species × CS × D, df = 3)
on the median proportional contributions from different soil depths to water uptake (MPC) of pea
and barley before the drought treatment on 7 May and the end of treatment on 25 June in 2018
tested by linear mixed models.

| Factor | Before drought treatment | | | End of drought treatment | | |
|---|---|---|---|---|---|---|
| | 0-20 cm | 20-40 cm | 40-60 cm | 0-20 cm | 20-40 cm | 40-60 cm |
| Species | **<0.001** | **0.036** | **<0.001** | **<0.001** | **<0.001** | **<0.001** |
| CS | 0.506 | 0.555 | 0.992 | 0.374 | 0.440 | 0.252 |
| D | 0.849 | 0.775 | 0.629 | **0.003** | 0.546 | **0.004** |
| Species × CS | 0.255 | 0.865 | 0.702 | 0.303 | 0.799 | 0.180 |
| Species × D | 0.424 | 0.619 | 0.336 | **0.009** | **0.001** | 0.359 |
| CS × D | 0.454 | 0.293 | 0.098 | 0.278 | 0.811 | 0.141 |
| Species × CS × D | 0.404 | 0.064 | 0.079 | 0.201 | 0.315 | 0.495 |
| Blocks | 0.360 | 0.667 | 0.534 | **0.008** | 0.115 | **0.016** |


MPC was derived from 10 000 simulations by mixing models using $\delta^{18}$O data. Proportional
contribution from 0-20 cm is the sum from 0-5, 5-10, and 10-20 cm, and 20-40 cm is the sum from
20-30 and 30-40 cm. CS and D were tested as two fixed effect factors for all subplots (*P* values are
given). Significant differences are shown in bold (*P* < 0.05).





Table 5 Effects of cropping systems (CS, df = 3), drought treatment (D, df = 1) and the interaction
(CS × D, df = 3) as well as the median proportional contributions from different soil depths to water
uptake (MPC) of pea and barley before the drought treatment on 7 May, at the end of treatment on
25 June, and after the drought treatment on 11 July in 2018 tested by linear mixed models.

| Species | Depth (cm) | P value from linear mixed models | | | | Mean ± 1 SE | | | | | | | |
|---|---|---|---|---|---|---|---|---|---|---|---|---|---|
| | | | | | | Control | | | | Drought | | | |
| | | CS | D | CS × D | Blocks | C-IT | C-NT | O-IT | O-RT | C-IT | C-NT | O-IT | O-RT |
| | | | | | | Before drought treatment | | | | | | | |
| Pea | 0-20 | 0.823 | 0.818 | 0.313 | 0.780 | 45±8 | 46±9 | 50±6 | 48±5 | 47±9 AB | 54±7 B | 34±9 A | 50±4 AB |
| | 20-40 | 0.557 | 0.834 | 0.913 | 0.656 | 37±6 | 29±3 | 32±4 | 36±3 | 35±7 | 27±3 | 36±8 | 33±3 |
| | 40-60 | 0.746 | 0.665 | 0.216 | 0.545 | 16±3 | 20±8 | 12±2 | 13±2 | 14±3 | 14±4 | 26±11 | 13±1 |
| Barley | 0-20 | 0.302 | 0.475 | 0.535 | 0.058 | 10±3 | 26±12 | 17±9 | 14±5 | 25±11 AB | 30±11 B | 11±7 A | 14±6 AB |
| | 20-40 | 0.736 | 0.707 | 0.156 | 0.785 | 41±16 ab | 39±9 ab | 65±16 b | 22±11 a | 55±15 AB | 29±10 A | 37±21 AB | 63±13 B |
| | 40-60 | 0.940 | 0.467 | 0.100 | 0.634 | 49±19 ab | 31±12 ab | 15±7 a | 63±17 b | 18±6 | 38±19 | 49±24 | 20±8 |
| | | | | | | End of drought treatment | | | | | | | |
| Pea | 0-20 | 0.416 | **0.001** | 0.17 | **0.010** | 63±6 b | 46±13 a | 48±9 ab | 51±4 ab | 65±4 AB | 64±14 A | 77±12 B | 64±7 A |
| | 20-40 | 0.416 | **0.003** | 0.703 | **0.021** | 27±5 a | 36±9 b | 31±3 ab | 31±1 ab | 23±2 AB | 24±9 B | 15±8 A | 24±5 B |
| | 40-60 | 0.398 | **0.008** | 0.272 | **0.027** | 8±1 a | 16±4 ab | 18±6 b | 14±5 ab | 9±1 AB | 10±4 B | 6±3 A | 8±1 AB |
| Barley | 0-20 | 0.214 | 0.459 | 0.488 | **0.034** | 43±2 | 38±11 | 28±6 | 30±8 | 47±7 B | 32±5 A | 31±9 AB | 39±7 AB |
| | 20-40 | 0.669 | 0.065 | 0.339 | 0.963 | 39±3 | 36±4 | 32±8 | 29±6 | 34±4 A | 47±4 B | 44±5 AB | 41±4 AB |
| | 40-60 | 0.207 | **0.017** | 0.213 | **0.028** | 15±1 a | 23±9 ab | 40±13 b | 38±13 b | 15±2 | 19±3 | 24±8 | 17±3 |
| | | | | | | After drought treatment | | | | | | | |
| Barley | 0-20 | 0.696 | 0.546 | 0.436 | **0.001** | 61±9 | 62±8 | 56±8 | 56±8 | 64±13 | 55±13 | 71±8 | 57±5 |
| | 20-40 | 0.664 | 0.604 | 0.508 | **0.004** | 28±6 | 25±5 | 30±5 | 31±5 | 25±9 | 31±9 | 20±6 | 31±4 |
| | 40-60 | 0.852 | 0.401 | 0.225 | **<0.001** | 10±2 | 11±3 | 13±4 | 12±3 | 11±4 | 13±4 | 7±2 | 10±1 |


MPC was derived from 10 000 simulations by mixing models using δ$^{18}$O data. Pea plants were
already senesced in early July therefore no stable water isotope data are available after the
treatment. Proportional contribution from 0-20 cm is the sum from 0-5, 5-10, and 10-20 cm, and 20-
40 cm is the sum from 20-30 and 30-40 cm. CS and D were tested as two fixed effect factors for all
subplots (P values are given). Significant differences are shown in bold (P < 0.05). Mean ± 1 SE for
MPC (%) are given for different cropping systems (C-IT for Conventional intensive tillage, C-NT
for Conventional no tillage, O-IT for Organic intensive tillage, and O-RT for Organic reduced
tillage). Different small and capital letters indicate significant differences among cropping systems
in control and drought subplots, respectively, tested with Tukey HSD (honestly significant
difference, P < 0.05).





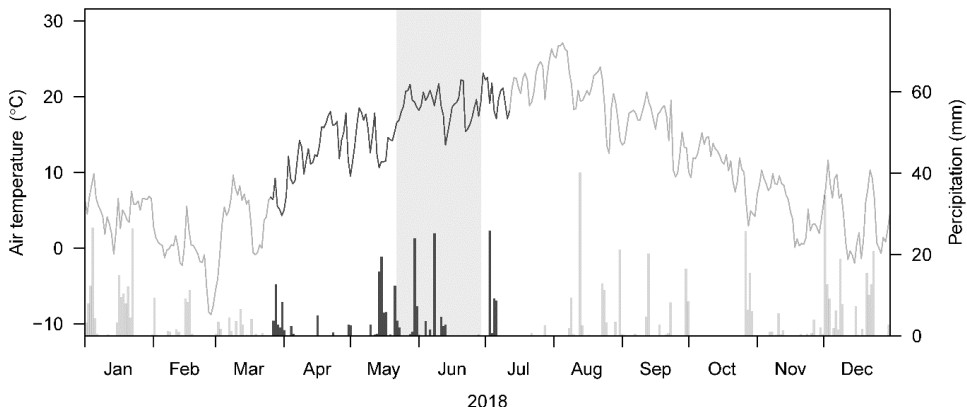


Fig. 1 Daily air temperature and precipitation in 2018. Dark line segments and bars depict the crop
growing season from 26 March to 12 July 2018. The shaded area indicates the drought treatment
from 22 May to 28 June 2018. Data from the MeteoSwiss station Zürich/Kloten (KLO, 47.48° N,
8.54° E, 4.6 km north of the research site, MeteoSwiss, 2020) are given.



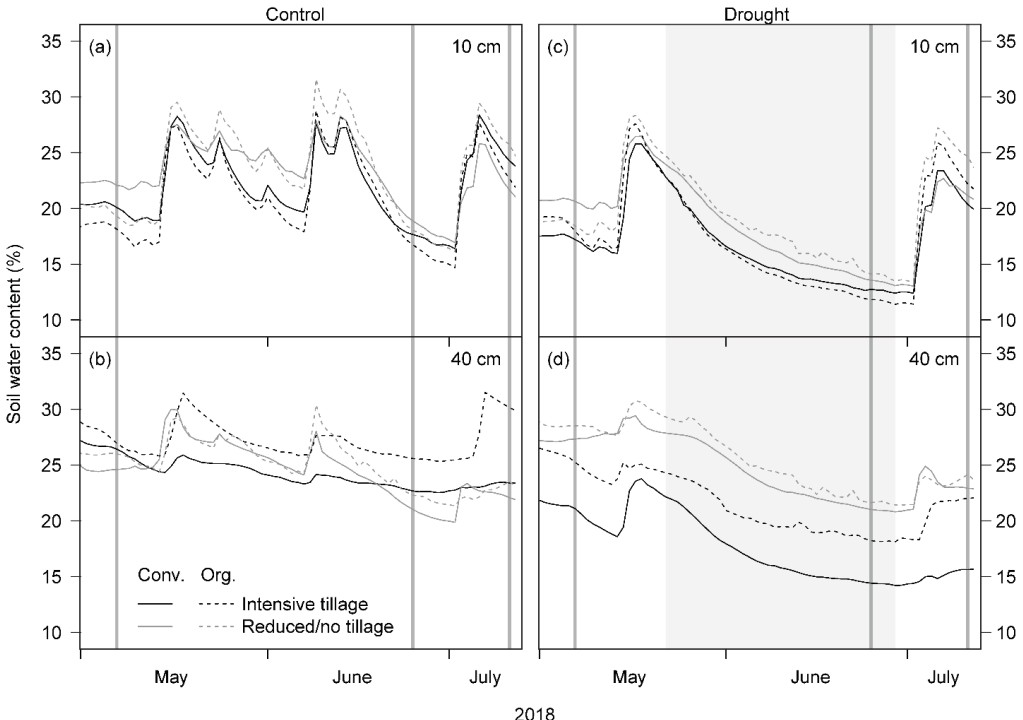


Fig. 2 Daily mean soil water contents at 10 and 40 cm depth in (a, b) control and (c, d) drought
subplots under different cropping systems (n = 2 each; Conv. for conventional, Org. for organic).
Vertical lines indicate sampling dates for stable water isotopes on 7 May, 25 June, and 11 July
2018. Shaded areas in (c) and (d) represent the drought treatment period from 22 May to 28 June

702    2018.



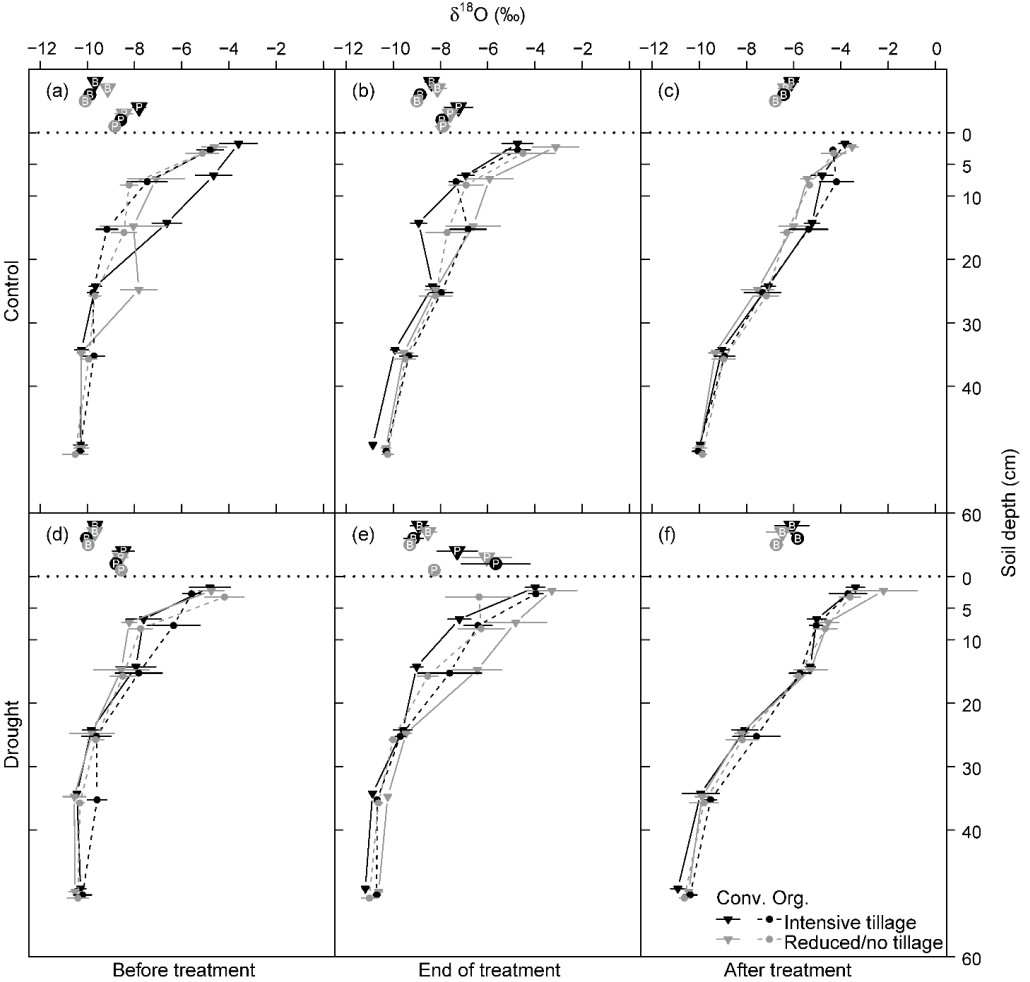

Fig. 3 $\delta^{18}O$ values of soil water from different depths and plant xylem water in each cropping
system (a, d) before the drought treatment on 7 May, (b, e) at the end of the drought treatment on 25
June, and (c, f) after treatment on 11 July in 2018 (Conv. for conventional, Org. for organic).
Horizontal dotted lines separate isotopic composition of soil and plant samples (P for pea, B for
barley). Pea plants were already senesced in early July, therefore no stable water isotope data are
available after the drought treatment. Means and 1 SE (horizontal bars) are given for each cropping
system (n = 3-4).

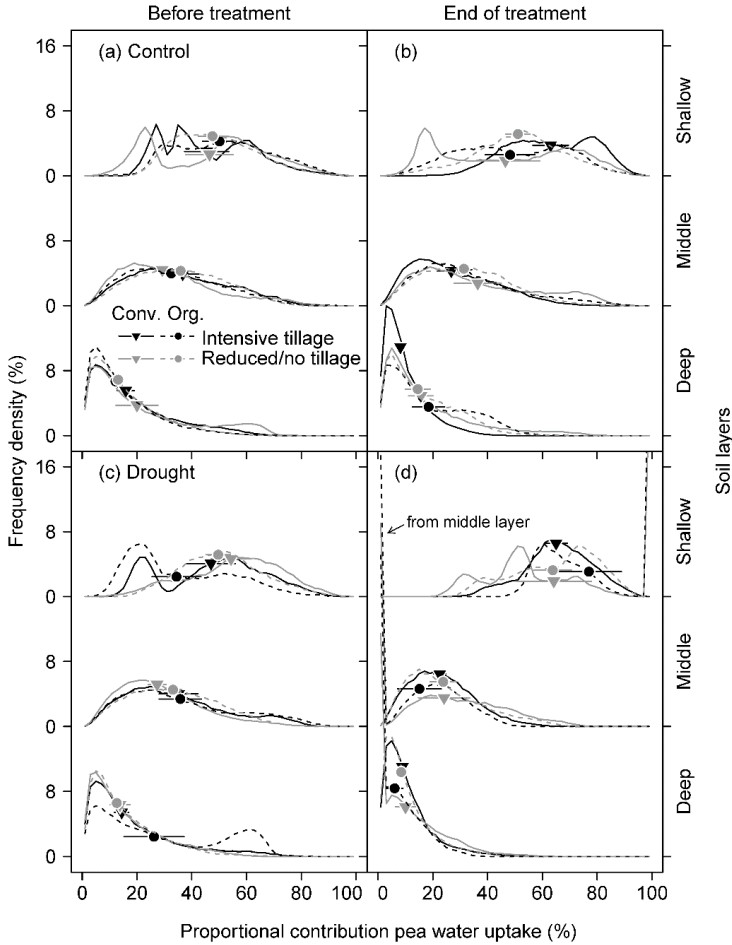

711

Fig. 4 Frequency density distribution of model outputs on the proportional contribution of soil water to pea water uptake from shallow (0-20 cm, sum of 0-5, 5-10, and 10-20 cm), middle (20-40 cm, sum of 20-30 and 30-40 cm), and deep (40-60 cm) soil layers under different cropping systems (a, b) before the drought treatment on 7 May and (c, d) at the end of treatment on 25 June in 2018. Frequency density was derived from 10 000 simulations at 2% increment of mixing models using $\delta^{18}O$ for each subplot (Conv. for conventional, Org. for organic). Data were pooled for all subplots in each cropping system. Symbols on the curves indicate the median of the model outputs for each soil layer. Means and 1 SE (horizontal bars) of each cropping system are given (n = 3-4).


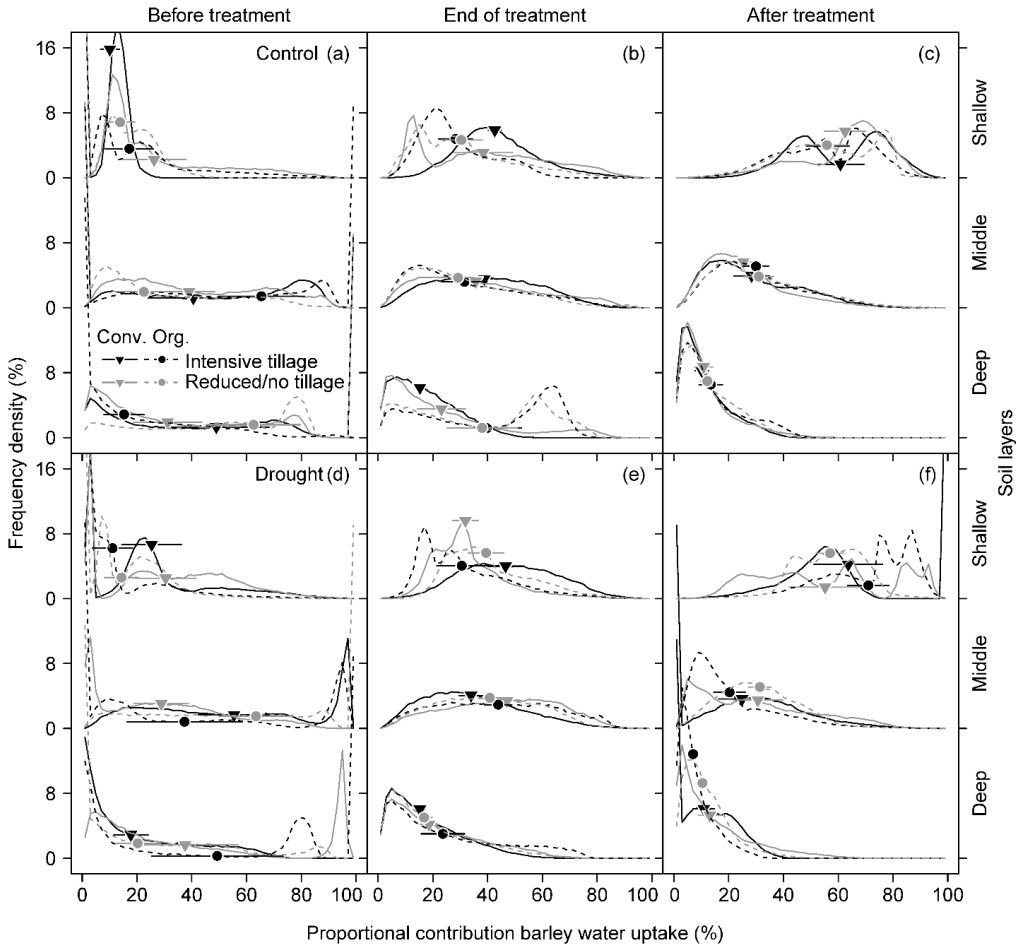


Fig. 5 Frequency density distribution of model outputs on the proportional contribution of soil water
to barley water uptake from shallow (0-20 cm, sum of 0-5, 5-10, and 10-20 cm), middle (20-40 cm,
sum of 20-30 and 30-40 cm), and deep (40-60 cm) soil layers under different cropping systems (a,
b) before the drought treatment on 7 May, (c, d) at the end of treatment on 25 June, and (e, f) after
treatment on 11 July in 2018. Frequency density was derived from 10 000 simulations at 2%
increment of mixing models using $\delta^{18}O$ for each subplot (Conv. for conventional, Org. for organic).
Data were pooled for all subplots in each cropping system. Symbols on the curves indicate the
median of the model outputs for each soil layer. Means and 1 SE (horizontal bars) of each cropping
system are given (n = 3-4).



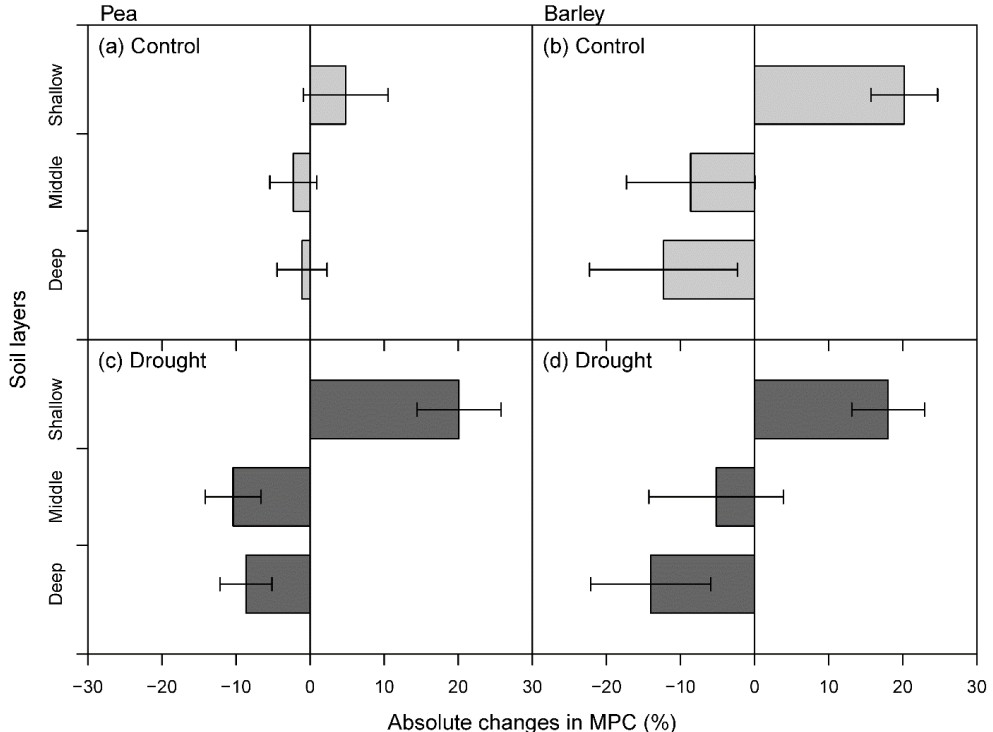


Fig. 6 Absolute changes in median proportional contributions to plant water uptake (MPC) of pea
(a, c) and barley (b, d), calculated as the difference of MPC at the end (25 June; ET) and before the
drought treatment (7 May; BT), from three soil layers in control (a, b) and drought (c, d) subplots.
MPC was derived from 10 000 simulations of mixing models using stable water isotope data.
Proportional contribution from the shallow layer is the sum of 0-5, 5-10, and 10-20 cm depths, the
middle layer is the sum of 20-30 and 30-40 cm depths, and the deep layer represents 40-60 cm.
Means and 1 SE (horizontal lines) are given (n = 14-16).

740