# Peer review of "Water uptake patterns of pea and barley responded to drought but not to cropping systems"

_Biogeosciences, 2021_

## Author Comment (AC1)

We thank the reviewer for their critical assessment and will provide a revised manuscript addressing the reviewer's comments. Throughout the following document, the reviewer's comment is stated first, followed by our response in *italic* font.

Comment on bg-2021-217

Anonymous Referee #1

Referee comment on "Water uptake patterns of pea and barley responded to drought but not to cropping systems" by Qing Sun et al., Biogeosciences Discuss., https://doi.org/10.5194/bg-2021-217-RC1, 2021

The manuscript by Qing Sun and colleagues does certainly address a topic within the scope of the journal. The authors used water stable isotopes to determine the effect of limited water availability on water uptake of a mixture of pea and barley. The data are certainly interesting and contribute to shedding some light on the still poorly understood role of soil management in coping with the consequences of climate change. Overall, I find it a good study, well structured, and clearly written.

- *We thank the reviewer for these kind words and positive feedbacks.*

I have a few comments, mostly minor.
Among the keywords there is "FAST": I do not understand what it means and how it should be a keyword for this paper.

- *FAST is the name of the long-term field trail we conducted the experiment in, given in the MatMet. We fully understand it is not an internationally well recognized name, therefore we will remove it from the keywords.*

The introduction is not particularly fluent. For example:
Row 38 – more frequent and more severe than what?

- *The original text is "Due to climate change, drought events may occur more frequently and become more severe." We will add "**than at present**" in the end. In addition, we will check the rest of the text carefully.*

Row 40 – Aggravating respect to what?

- *The original text is "Thus, agriculture is facing increasing pressure to ensure food security under aggravating conditions." We will change it into "aggravating **drought** conditions".*

Row 42 – the word "adapted" is superfluous and misleading (not clearly explained adapted to what)

- *Thank you for the comment. We will change it into "**Potentially adaptive** crop management to the changing climate…"*

Row 60 – "illustrating the current gap of knowledge for cropping systems" not sure what does this mean.

- *We agree that the wording was not explicit enough. We will change this to "gap of knowledge **on plant water relations in** cropping systems".*

Regarding the methods:

Row 85: reporting also the altitude of the experiment would be useful.

- *Thank you for the suggestion. We will add "**489 m a.s.l.**" in the text.*

Row 106: what was the proportion of pea and barley in the mixture?

- *The mixture was composed of 20% and 80% of the recommended sowing densities of pea (90 seeds/m$^2$) and barley (350 grains/m$^2$), respectively. The seeds were sowed in a mixture with a standard drill-sowing machine. We will add this information in the text.*

Row 142: a short explanation of how cryogenic vacuum distillation was performed would be advisable.

- *Thank you for the suggestion. We will add "**During the extraction, the samples were kept in an 80 •C water bath, extracted under 10$^{-2}$ MPa for 2 h, and collected in liquid nitrogen**."*

Row 145: which peripheral was used?

- *TC/EA was used. We will revise the sentence accordingly as the following: "The oxygen and hydrogen stable isotope ratios ($\delta^{18}O$ and $\delta^{2}H$) of water samples were analysed by coupling a high-temperature elemental analyser (TC/EA, Finnigan MAT, Bremen, Germany) with an isotope ratio mass spectrometer (IRMS, Delta$^{plus}$XP, Finnigan MAT, Bremen, Germany) via a ConFlo III interface (Finnigan MAT, see Werner et al., 1999), using the high-temperature carbon reduction method described by Gehre et al., (2004)."*

Row 146: I do not understand the citation of Werner et al 1999 since this paper describes the analysis of nitrogen and carbon stable isotopes. Please clarify or change with a more appropriate citation.

- *Thanks for the suggestion, our referral was not clear. We will replace Werner et al 1999 with "**Gehre M, Geilmann H, Richter J, Werner RA, Brand WA. 2004.** Continuous flow $^{2}H/^{1}H$ and $^{18}O/^{16}O$ analysis of water samples with dual inlet precision. Rapid Communications in Mass Spectrometry **18**(22): 2650-2660."*

Row 180: I'd suggest modifying the naming of the treatments. The term "after treatment" is (to me) a little confusing because the treatment had already been finished for a couple of weeks at the moment of sampling. In the M&M section, it is explained that the terms refer to the phases, but, still, it is confusing. For example, in table 1 it is reported a period "before drought", which makes sense, and "after drought", which also makes sense, but also a period "end of drought", which does not make sense since this is the period during which the drought actually took place! Maybe it would be better to call the last sampling something like "after rain" or "end of experiment" or simply "last sampling".

- *Thank you for the suggestion. We wanted to emphasise the timepoint for the sampling campaigns relative to the presence of the shelters, which simulated an experimental drought. The term "end of drought" fits better to the treatment duration and indicates that the sampling actually did not take place "after" the drought treatment, but shortly before the treatment was ended, i.e., before the shelters were removed. Since our drought study was an experiment, replacing "drought" with "experiment" would not be correct, since our study (i.e., our experiment) continued even after the drought simulation had finished (i.e., shelters removed). With the term "end of drought" we want to clarify that the plots were subjected by the drought for some time before our sampling took*

*place. Therefore, after discussing this issue among the co-authors, we would prefer to keep the terms as they are defined in the Materials and Methods section.*

Results

Row 210-211: also the fact that soil water content in the low tillage plots at 40 cm was higher in the drought plots than in the "control" plots should be highlighted and discussed

- *Since we had only two replicates of soil water content measurements, we did not conduct statistical analysis on the data, but used the data rather qualitatively, as differences between cropping systems could not be tested. Although the soil water content in systems with conservation tillage was slightly higher in drought than in control subplots since before the drought treatment (28 vs 27% in C-NT and 29 vs 27% in O-RT), it continuously decreased during the treatment in the drought subplot, while it responded to rain and natural conditions in the control subplots. Therefore, we did not emphasise the absolute differences among cropping systems or between treatments, but rather showed the changes of soil water content over time and how they were affected by the drought treatment. The slight differences in soil water content among cropping systems were covered in discussion (row 399-409). We will make sure this aspect is better explained in the final version of this manuscript.*

Row 213-218: I do not understand the relevance of reporting the soil and xylem water results with respect to the LMWL since this aspect is not further discussed. It might be of certain general interest to see if there is any difference due to the drought, but it is my understanding that the data in figure S1 are reported for the whole vegetative season.

- *Thank you for the comment. It is mainly to show the data points fall in a reasonable range. But that is a good point. We will separate this figure and show the three sampling campaigns in a revised version.*

Discussion

The main point that, in my opinion, is missing from the discussion is considering the type of soil. I understand that different species may act differently because of their genotype, but soil type (texture, bulk density, and soil organic matter content in particular) has a huge influence on soil water fluxes and root distribution (of any species). In addition, the soil type may influence the effect of soil management on plants' behavior, water uptake in this case.

- *Thank you for the suggestion. In general, soil type indeed would be an important aspect to discuss to explain water uptake depths as mentioned in the introduction (row 42-48 and 63-65) and the discussion (row 392-409). However, at our site, the soil type did not change across the field site, and the experimental set-up with a block design accounted for spatial variations. Although Wittwer et al. (2021) showed differences in soil properties in top 20 cm (e.g., soil carbon to clay ratio, aggregate mean weight diameter), our data showed no significant cropping system effects nor interactions between drought and cropping system effects on root water uptake patterns. Thus, soil physical properties among cropping systems cannot provide additional information on the similar drought responses observed on both pea and barley plants in water uptake patterns. We will make this clearer in a revised version of the final manuscript.*
*Wittwer RA, Bender SF, Hartman K, Hydbom S, Lima RAA, Loaiza V, Nemecek T, Oehl F, Olsson PA, Petchey O, et al. 2021. Organic and conservation agriculture promote ecosystem multifunctionality. Science Advances 7(34).*

Other comments

Row 340 – but in the low tillage plots soil moisture at 40 cm is higher in the drought treatment than in the control. How do you justify this behavior?

- *Please refer to our answer above. With two sensors per cropping system, we cannot test for significant differences and therefore hesitated to put a lot of emphasis on the absolute soil moisture values, rather using their temporal development into the growing season/drought treatment. In addition, the absolute differences were rather small, particularly compared to the natural variability observed.*

Row 355 – do you think that plants had enough time during the drought period to adapt their root distribution?

- *Yes, we think so. The 5-week drought treatment started at an early stage of the vegetative phase when both crops, pea and barley plants, were around 20 to 30 cm tall. At the end of drought treatment, the plants reached around 60 cm. In parallel to above-ground growth, also belowground biomass is grown to keep the root:shoot ratio in balance. Thus, although we do not have root growth data, we still see changes in water uptake depths for the species. But in general, annual crops with a short vegetation period, have a rather short window of opportunity to acclimate their root distributions. The period of 5 weeks of no/less rain is a fairly strong treatment, comparable to what we expect in the future. In addition, from what we observed in grassland species, grasses increased carbon allocation to roots and more so in shallow soil layer under drought conditions (see Prechsl et al., 2015, Burri et al., 2014). We will follow up this line of argument in a revised version.*

  Prechsl UE, Burri S, Gilgen AK, Kahmen A, Buchmann N. 2015. No shift to a deeper water uptake depth in response to summer drought of two lowland and sub-alpine C₃-grasslands in Switzerland. Oecologia 177(1): 97-111.

  Burri S, Sturm P, Prechsl UE, Knohl A, Buchmann N. 2014. The impact of extreme summer drought on the short-term carbon coupling of photosynthesis to soil $CO_2$ efflux in a temperate grassland. Biogeosciences 11(4): 961-975.

Row 367 – table 5 does not show the water uptake during the "natural dry" (I suggest changing to "naturally dry") period.

- *Thank you for the suggestion. We will revise it accordingly.*

row 389 – what does "_ENREF_5" mean?

- *We apologize. That is a broken link which was previously removed. It will be fixed.*

Row 403-404 – in stating that there is no difference in MPC due to the cropping system, have you considered the difference between before and end of treatment? By doing a very very rough calculation, it seems to me that there might be some influence of the treatment at least in peas.

- *Thanks for the suggestion. Yes, we have calculated the difference between before and end of the treatment for different cropping systems and found no significant effects of cropping systems. Thus, we did not include this info in the original version. However, we will include these results in supporting documents as Table. S6 and refer to them in the text from row 316.*

Tables and figures

**Commented [SQ1]:** By Anna and Nina: We could add that we know from grassland that the response in root growth is really fast. Prechsl, also Burri

Table 5 – the legend is not clear: the effect on what? [I guess it is on the MPC, but it is not clear from the legend]

- *We will revise the caption accordingly.*

Table 5 is very dense and difficult to follow. It would be easier to read if the letters were closer to the respective number. Or consider splitting in two.

- *We will re-format the table to make it clearer.*

Fig. 6 – are these data pooled for all soil managements? Please specify in the legend.

- *Thanks for the suggestion. We will add "**in all cropping systems**" in the caption.*

---

## Author Comment (AC2)

We thank the reviewer for their critical assessment and will provide a revised manuscript addressing the reviewer's comments. Throughout the following document, the reviewer's comment is stated first, followed by our response in *italic* font.

Comment on bg-2021-217
Anonymous Referee #2

Referee comment on "Water uptake patterns of pea and barley responded to drought but not to cropping systems" by Qing Sun et al., Biogeosciences Discuss., https://doi.org/10.5194/bg-2021-217-RC2, 2022

General comments:    Adaptive crop management is a potentially important strategy to mitigate crop failures due to climate-change induced drought stress. How plants utilize water across different cropping systems could determine the extent to which a particular management system can be used to mitigate plant drought stress. The authors argue that very few studies have evaluated plant water uptake patterns in response to drought in arable cropping systems. The manuscript is generally well written, and I have mostly minor suggestions for improving clarity in a few places.

- *Thank you very much for this very positive evaluation.*

(1) My one main concern about the study is that it simulates a very short-term drought during a natural drought, such that moisture conditions in the drought treatments were not that different from control plots. The authors should provide a stronger rationale for why the results of such a short-term study are relevant. (2) That is, why would differences between cropping systems be expected given the experimental design and natural drought conditions during the study period? (3) The authors should also discuss if/why one year of data collection is sufficient, especially given that a natural drought occurred during the study period. Are there data available from previous or subsequent years that could be brought to bear on these questions? For example, are there data on root biomass that could be included to strengthen the paper? (4) I would also like to see acknowlegment and discussion of the role of mycorrhizal fungi on drought responses.

- *Thank you very much for the comments. We added numbers (1-4) in the above comment to facilitate evaluation of our answers.*
- *(1) Regarding the length/severity of our drought treatment: Our simulated drought was not very short but actually 5 weeks long. Compared to our local climate conditions, and on the background of studying an annual crop species with a rather short growing phase, a 5-week drought is quite severe. This can also be seen in Fig. 1, where 5 weeks without precipitation were absent in 2018. We excluded 34% of the precipitation during the growing season, as given in Tab. 1, The scenarios for Switzerland (i.e., CH2018, https://www.meteoswiss.admin.ch/home/latest-news/news/climate-scenarios-ch2018.html; page 6) foresee a reduction up to 25% in precipitation in Switzerland in 2060 due to climate change, up to 40% by the end of the century. Our reduction for 2018 was right in the middle of these two numbers. In addition, the CH2018 scenarios foresee an increase of the longest rain-free summer period (June, July, August) from currently 11 days to 20 days, even shorter than our 5-week drought simulation. Thus, both length and severity of our treatment were very strong, clearly strong enough to simulate a future drought scenario threatening food security. We will add this information in the revised version.*

- *(2) Regarding the natural drought: The natural drought 2018 occurred end of June, at the end of our experimental drought treatment (see Figs. 1 and 2), when the plants had already received about 150 mm of rain (see Tab. 1) and most likely also developed their main rooting system according to non-drought conditions. We expected differences among cropping systems because the Farming Systems and Tillage Experiment was established in 2009, i.e., the cropping systems were already in place 9 years before our drought study took place. It has been reported that these cropping systems in our site resulted in different soil properties (see Wittwer et al., 2021). Wittwer RA, Bender SF, Hartman K, Hydbom S, Lima RAA, Loaiza V, Nemecek T, Oehl F, Olsson PA, Petchey O, et al. 2021. Organic and conservation agriculture promote ecosystem multifunctionality. Science Advances 7(34).*

- *(3) Regarding the number of study years, we think this one year of experiment with an intensive sampling and measurement campaign is sufficient to show the response of a short-term crop like pea and barley and their water uptake to drought and the lack of cropping systems effects.*
  - *Our experimental design included drought and control sub-plots. Thus, we worked with replicated subplots in parallel (i.e., at the same time), not after each other (i.e., a temporal replication over multiple years). Since we were interested in the response of pea and barley to our drought treatment and not in their long-term yields, we think that our experiment design is adequate.*
  - *In addition, in croplands, the identical crop should not be grown on the same field for several years due to arising soil health issues. This is the reason why agriculture in many countries works with rather elaborate crop rotation schemes, not only in Switzerland. Our study is no exception in this respect, a 6-yr rotation is used on this site and this rotation never has two legume crops following each other. So, it would not have been possible to again grow pea-barley in the following years on the same plots.*
  - *The increasing competition/lack of niche differentiation observed between pea and barley grown in a mixture under the drought treatment within the usual cropping season was surprising and deserves attention since farmers need to know about such an undesired outcome.*
  - *Besides, the natural drought was not a disruption for the experiment. On the contrary, with the natural drought (in control subplots, occurring at the end of the drought treatment) and the simulated drought (in drought subplots), we could study an aggravated drought scenario, one that is becoming more likely in the future.*

> **Commented [KV1]:** Does this still refer to (3)? I feel the arguments before are sufficient and fit better to what the reviewer asks
>
> **Commented [SQ2R1]:** It's about the "especially there was a natural drought.."

- *(4) Regarding mycorrhizal fungi: We agree that mycorrhization is worth mentioning as one of the reasons that can cause different plant water uptake patterns in cropping systems. Some studies discussed the positive effects of organic management and possibly from conservation tillage, maybe modulated via mycorrhizal fungi. Therefore, we will address this aspect in the introduction. Our results clearly showed no interactions between cropping systems and the simulated drought; and the potential benefits from soil as affected by different management practices were not able to overcome the drought effects (L399-409). So we will also address mycorrhizal activities as part of those soil functions that were overwhelmed by the simulated drought.*

Specific comments:

L13-17. Some of the Highlights bullets are unclear as written

- *Thank you for the comments. We guess that the lack of "was" or "were" created this impression. We will rephrase.*

L22-23. Which species is the important "fodder crop" or is it the mixture that is the "crop".

- *The 'pea-barley mixture' as an intercrop is the important fodder crop. We will clarify this in a revised version.*

L29-30. Check sentence construction.

- *We will revise it as "Both species showed similar responses to the drought simulation and increased their proportional **water uptake** from shallow soil layer (0-20 cm) in all cropping systems."*

L42. "Adapted" should read "Adaptive"

- *Thank you for the suggestion.*

L63-70. Root-mycorrhizal interactions are also presumably important, but are not discussed here (and should be)

- *Thank you for the suggestion. We will include references discussing the potential benefits on mycorrhiza in organic systems. See also response (4) above.*

L80-81. As opposed to when grown alone?

- *No, this is a misunderstanding. Just to compare the two species in the mixture. We will check and improve the sentence when revising the document.*

L111-112. What are typical rainfall patterns during this period? Is a one-month drought during this time atypical? This seems like a fairly short period of drought conditions. Also, what was the antecedent conditions in the months preceding the drought period. If rainfall was at or above average, perhaps a one month drought is not a problem? If I'm interpreting Table 1 correctly, it actually looks like precipitation in the control plots was ~30% of normal, so it appears there was a natural drought during the imposed drought period. I wonder why water wasn't added to control plots to simulate a "normal" year.

- *Thank you for the questions.*
  - o *Our study did not aim at an atypical or typical drought, but at a prolonged drought as predicted under climate change; the drought treatment was more than a month (37 d) and is therefore substantial for a three-and-a-half-month growing season for our pea-barley mixture. Currently, the longest period of no rain during summer is 11 days. Our 37-day drought is therefore quite long.*
  - o *Indeed, there was a natural drought, which was also addressed in discussion (L361-366) and can be seen in Tab. 1 as well as in Figs 1 and 2. In Tab. 1, one can also see that May precipitation was similar to the long-term mean (102 vs 105 mm). So no water surplus or scarcity before our drought treatment. We will describe Tab. 1 better in the MatMet section to avoid any misunderstanding here.*
  - o *We did not irrigate the control subplot during the natural drought for different reasons: (i) We did not want to simulate a "normal" year vs. a drought period, but add the drought stress on*

*top of the actual rainfed conditions. Also because irrigation is no typical agricultural measure in this region and the natural drought was not extraordinary strong so that it would have caused substation yield losses. (ii) We clearly did not anticipate such a natural drought during our experiment. Adding irrigation short-term was beyond our logistic possibilities and irrigating the control subplots with tap water (since no rainwater could have been collected during the natural drought) would have added nutrients to the controls and thus would have altered soil chemistry. This would have destroyed our experimental design.*

L123-128. Describe antecedent moisture conditions here (i.e., per my questions above).

- *Thank you for the suggestion. However, we do not fully understand what is requested here in addition to what we already provide. In Tab. 1, we give the long-term climatic conditions as well as those during the growing season 2018 and separate months. In Figs 1 and 2 we show the temperature and the precipitation during the entire year 2018 (precipitation; Fig. 1) and during our experiment (soil moisture; Fig. 2). So we think we have covered the antecedent moisture conditions already.*

L322-325. I don't understand this response. Under drought, when the surface soil is presumably dryer, wouldn't plants tend to have deeper rooting depths and/or obtain more water from depth?

- *Thank you for the question. Indeed, this question is what the Bayesian mixing models are used for, based on the stable water isotope data. While deeper rooting in response to drought is the usual assumption, experiments and observations do not always show such reliance on deeper soil layers during drought. We discussed such results in comparison to ours in detail in discussion (L326-341 and L342-360).*

L415. "adaptive" not "adapted"

- *Thank you for the correction. We will change the word accordingly.*

L415-417. I would take some care with this conclusion, acknowledging that cropping system differences may be more pronounced under longer-term drought (see previous comments above)

- *Thank you for the suggestion. As you have pointed out above, in addition to our drought treatment, there was a natural drought (leading to only 40% of monthly precipitation in June 2018 compared to long-term average in June under rainfed conditions). Our 37-d drought treatment simulated a much more severe drought than the natural drought (excluding 79 mm). Altogether, we report data of a "more pronounced drought". Still, we did not find significant differences in changing water uptake patterns among the different cropping systems. We consider our results sufficient to conclude that under future climate with similarly small water supply, cropping systems might not be reliable to provide consistent alleviation for plant drought stress for pea-barley. But as also written, it remains to be seen how other crops react.*

Table 1. It unclear what the last three lines in the table refer to. Are these precipitation amounts in the drought treatment itself?

- *Almost. As indicated in the first column, the values are total precipitation and mean air temperature during the respective time periods separated by the drought treatment. Therefore, not just in the drought treatment itself, but before, during, and after the drought treatment. We will revise the second last cell (change "End of drought treatment" to "During drought treatment").*

Table 2-5. I assume the values shown are P-values, but that is not explicilty stated in the table heading. This is a lot of tables of p values...

- *Thank you for the comment. Yes, they are p values. We will revise the captions to clarify the values.*

L671-674. I'm unclear why these lines are shown below the table. Is this text part of the heading or a footnote to the table?

- *It serves as a footnote to the table.*

---

## Author Response (AR1)

We thank the editor for the positive feedbacks and the reviewers for their critical assessment. Here we provide a detailed response addressing the reviewers' comments. Throughout the following document, the reviewers' comments are stated first, followed by our response in *italic* font. As required, the revised manuscript has track-changes, therefore line numbers in the response refer to tracking mode with all markup.

Comment on bg-2021-217

Anonymous Referee #1

Referee comment on "Water uptake patterns of pea and barley responded to drought but not to cropping systems" by Qing Sun et al., Biogeosciences Discuss., https://doi.org/10.5194/bg-2021-217-RC1, 2021

The manuscript by Qing Sun and colleagues does certainly address a topic within the scope of the journal. The authors used water stable isotopes to determine the effect of limited water availability on water uptake of a mixture of pea and barley. The data are certainly interesting and contribute to shedding some light on the still poorly understood role of soil management in coping with the consequences of climate change. Overall, I find it a good study, well structured, and clearly written.

- *We thank the reviewer for these kind words and positive feedbacks.*

I have a few comments, mostly minor.

Among the keywords there is "FAST": I do not understand what it means and how it should be a keyword for this paper.

- *FAST is the name of the long-term field trail we conducted the experiment in, given in the MatMet (L92). We fully understand it is not an internationally well recognized name, therefore we removed it from the keywords.*

The introduction is not particularly fluent. For example:

Row 38 – more frequent and more severe than what?

- *The original text was "Due to climate change, drought events may occur more frequently and become more severe." We added "**than at present**" in the end (L38-39).*

Row 40 – Aggravating respect to what?

- *The original text was "Thus, agriculture is facing increasing pressure to ensure food security under aggravating conditions." We changed it into "aggravating **drought** conditions" (L41).*

Row 42 – the word "adapted" is superfluous and misleading (not clearly explained adapted to what)

- *Thank you for the comment. We changed it into "**Adaptive** crop management **to a changing climate** ..." (L43).*

Row 60 – "illustrating the current gap of knowledge for cropping systems" not sure what does this mean.

- *We agree that the wording was not explicit enough. We changed this to "gap of knowledge **on plant water relations in** cropping systems" (L61).*

Regarding the methods:

Row 85: reporting also the altitude of the experiment would be useful.

- *Thank you for the suggestion. We added "**489 m a.s.l.**" in the text (L86).*

Row 106: what was the proportion of pea and barley in the mixture?

- *The mixture was composed of 20% and 80% of the recommended sowing densities of pea (90 seeds/$m^2$) and barley (350 grains/$m^2$), respectively. The seeds were sown in a mixture with a standard drill-sowing machine. We added this information in the text (L108-111).*

Row 142: a short explanation of how cryogenic vacuum distillation was performed would be advisable.

- *Thank you for the suggestion. We added "**During the extraction, the samples were kept in an 80 °C water bath, extracted under $10^{-2}$ MPa for 2 h, and the extracted water collected in glass tubes immersed in liquid nitrogen**." (L156-157)*

Row 145: which peripheral was used?

- *A TC/EA was used. We revised the sentence accordingly as the following: "The oxygen and hydrogen stable isotope ratios ($\delta^{18}O$ and $\delta^2H$) of water samples were analysed by coupling a high-temperature elemental analyser (TC/EA, Finnigan MAT, Bremen, Germany) with an isotope ratio mass spectrometer (IRMS, Delta$^{plus}$XP, Finnigan MAT, Bremen, Germany) via a ConFlo III interface (Finnigan MAT, see Werner et al., 1999), using the high-temperature carbon reduction method described by Gehre et al., (2004)." (L159-163)*

Row 146: I do not understand the citation of Werner et al 1999 since this paper describes the analysis of nitrogen and carbon stable isotopes. Please clarify or change with a more appropriate citation.

- *Thanks for the suggestion, our referral was not clear. The Werner et al. 1999 reference describes the ConFlo interface. We changed the sentence and replaced this reference with "**Gehre M, Geilmann H, Richter J, Werner RA, Brand WA. 2004.** Continuous flow $^2H/^1H$ and $^{18}O/^{16}O$ analysis of water samples with dual inlet precision. Rapid Communications in Mass Spectrometry **18**(22): 2650-2660." (L163)*

Row 180: I'd suggest modifying the naming of the treatments. The term "after treatment" is (to me) a little confusing because the treatment had already been finished for a couple of weeks at the moment of sampling. In the M&M section, it is explained that the terms refer to the phases, but, still, it is confusing. For example, in table 1 it is reported a period "before drought", which makes sense, and "after drought", which also makes sense, but also a period "end of drought", which does not make sense since this is the period during which the drought actually took place! Maybe it would be better to call the last sampling something like "after rain" or "end of experiment" or simply "last sampling".

- *Thank you for the suggestion. We wanted to emphasise the timepoint for the sampling campaigns relative to the presence of the shelters, which simulated an experimental drought. The term "end of drought" fits better to the treatment duration and indicates that the sampling actually did not take place "after" the drought treatment, but shortly before the treatment was ended, i.e., before the shelters were removed. Since our drought study was an experiment, replacing "drought" with "experiment" would not be correct, since our study (i.e., our experiment) continued even after the*

*drought simulation had finished (i.e., shelters removed). With the term "end of drought" we want to clarify that the plots were subjected to the drought for some time before our sampling took place. Therefore, after discussing this issue among the co-authors, we would prefer to keep the terms as they are defined in the MatMet section (L143-144).*

Results

Row 210-211: also the fact that soil water content in the low tillage plots at 40 cm was higher in the drought plots than in the "control" plots should be highlighted and discussed

- *Since we had only two replicates of soil water content measurements, we did not conduct statistical analysis on the data, but used the data rather qualitatively, as differences between cropping systems could not be tested. Although the soil water content in systems with conservation tillage was slightly higher in drought than in control subplots before the drought treatment (28 vs 27% in C-NT and 29 vs 27% in O-RT; Fig. 2c and d), it continuously decreased during the treatment in the drought subplot (Fig. 2d), while it responded to rain and natural conditions in the control subplots (Fig. 2c). Therefore, we did not emphasise the absolute differences among cropping systems or between treatments, but rather showed the changes of soil water content over time and how they were affected by the drought treatment. The slight differences in soil water content among cropping systems were also covered in discussion (L430-437).*

Row 213-218: I do not understand the relevance of reporting the soil and xylem water results with respect to the LMWL since this aspect is not further discussed. It might be of certain general interest to see if there is any difference due to the drought, but it is my understanding that the data in figure S1 are reported for the whole vegetative season.

- *Thank you for the comment. It is mainly to show the data points fall in a reasonable range. But that is a good point. We separated this figure and now show the three sampling campaigns separately in the revised version (Fig. S1).*

Discussion

The main point that, in my opinion, is missing from the discussion is considering the type of soil. I understand that different species may act differently because of their genotype, but soil type (texture, bulk density, and soil organic matter content in particular) has a huge influence on soil water fluxes and root distribution (of any species). In addition, the soil type may influence the effect of soil management on plants' behavior, water uptake in this case.

- *Thank you for the suggestion. In general, soil type indeed would be an important aspect to discuss to explain water uptake depths as mentioned in the introduction (L43-50 and L62-68) and the discussion (L423-442). However, at our site, the soil type did not change across the field site, and the experimental set-up with a block design accounted for spatial variations. Although Wittwer et al. (2021; now cited in L50) showed differences in soil properties in the top 20 cm (e.g., soil carbon to clay ratio, aggregate mean weight diameter) due to organic management and conservation tillage, our data showed no significant cropping system effects nor interactions between drought and cropping system effects on root water uptake patterns. Thus, soil physical properties among cropping systems cannot provide additional information on the similar drought responses observed on both pea and barley plants in water uptake patterns. We added a sentence to the discussion in L435-437.*
*Wittwer RA, Bender SF, Hartman K, Hydbom S, Lima RAA, Loaiza V, Nemecek T, Oehl F, Olsson*

*PA, Petchey O, et al. 2021. Organic and conservation agriculture promote ecosystem multifunctionality. Science Advances 7(34).*

Other comments

Row 340 – but in the low tillage plots soil moisture at 40 cm is higher in the drought treatment than in the control. How do you justify this behavior?

- *Please refer to our answer above. With two sensors per cropping system, we cannot test for significant differences and therefore hesitated to put a lot of emphasis on the absolute soil moisture values, rather using their temporal development into the growing season/drought treatment. In addition, the absolute differences were rather small, particularly compared to the natural variability observed.*

Row 355 – do you think that plants had enough time during the drought period to adapt their root distribution?

- *Yes, we think so. The 5-week drought treatment started at an early stage of the vegetative phase when both crops, pea and barley plants, were around 20 to 30 cm tall. At the end of drought treatment, the plants reached around 60 cm. In parallel to above-ground growth, also belowground biomass is grown to keep the root:shoot ratio in balance. Thus, although we do not have root growth data, we still see changes in water uptake depths for the species. But in general, annual crops with a short vegetation period, have a rather short window of opportunity to acclimate their root distributions. The period of 5 weeks of no/less rain is a fairly strong treatment, comparable to what we expect in the future. In addition, from what we observed in grassland species, grasses increased carbon allocation to roots and more so in shallow soil layer under drought conditions (e.g., see Prechsl et al., 2015 as cited in discussion, L374-375). We added some more info in this paragraph (L384-386).*
*Prechsl UE, Burri S, Gilgen AK, Kahmen A, Buchmann N. 2015. No shift to a deeper water uptake depth in response to summer drought of two lowland and sub-alpine $C_3$-grasslands in Switzerland. Oecologia 177(1): 97-111.*

Row 367 – table 5 does not show the water uptake during the "natural dry" (I suggest changing to "naturally dry") period.

- *Thank you for the suggestion. We revised the sentence accordingly to make it clearer (L398). In Table 5, the results on water uptake at the end of treatment for control subplots give information on the response to the naturally dry period.*

row 389 – what does "_ENREF_5" mean?

- *We apologize. That is a broken link which was previously removed. It is fixed now (L420).*

Row 403-404 – in stating that there is no difference in MPC due to the cropping system, have you considered the difference between before and end of treatment? By doing a very very rough calculation, it seems to me that there might be some influence of the treatment at least in peas.

- *Thanks for the suggestion. Yes, we have calculated the difference between before and end of the treatment for different cropping systems and found no significant effects of cropping systems. Thus, we did not include this info in the original version. However, we now included these results in supporting documents as Table S6 and referred to them in the text in L332-335.*

Tables and figures

Table 5 – the legend is not clear: the effect on what? [I guess it is on the MPC, but it is not clear from the legend]

- *We revised the caption accordingly.*

Table 5 is very dense and difficult to follow. It would be easier to read if the letters were closer to the respective number. Or consider splitting in two.

- *We re-formatted the table to make it clearer.*

Fig. 6 – are these data pooled for all soil managements? Please specify in the legend.

- *Thanks for the suggestion. We added "**in all cropping systems**" in the caption.*

Comment on bg-2021-217

Anonymous Referee #2

Referee comment on "Water uptake patterns of pea and barley responded to drought but not to cropping systems" by Qing Sun et al., Biogeosciences Discuss., https://doi.org/10.5194/bg-2021-217-RC2, 2022

General comments:    Adaptive crop management is a potentially important strategy to mitigate crop failures due to climate-change induced drought stress. How plants utilize water across different cropping systems could determine the extent to which a particular management system can be used to mitigate plant drought stress. The authors argue that very few studies have evaluated plant water uptake patterns in response to drought in arable cropping systems. The manuscript is generally well written, and I have mostly minor suggestions for improving clarity in a few places.

- *Thank you very much for this very positive evaluation.*

(1) My one main concern about the study is that it simulates a very short-term drought during a natural drought, such that moisture conditions in the drought treatments were not that different from control plots. The authors should provide a stronger rationale for why the results of such a short-term study are relevant. (2) That is, why would differences between cropping systems be expected given the experimental design and natural drought conditions during the study period? (3) The authors should also discuss if/why one year of data collection is sufficient, especially given that a natural drought occurred during the study period. Are there data available from previous or subsequent years that could be brought to bear on these questions? For example, are there data on root biomass that could be included to strengthen the paper? (4) I would also like to see references acknowlegment and discussion of the role of mycorrhizal fungi on drought responses.

- *Thank you very much for the comments. We added numbers (1-4) in the above comment to facilitate evaluation of our answers.*

- *(1) Regarding the length/severity of our drought treatment: Our simulated drought was not very short but actually 5 weeks long. Compared to our local climate conditions, and on the background of studying an annual crop species with a rather short growing phase, a 5-week drought is quite severe. This can also be seen in Fig. 1, where 5 weeks without precipitation were absent in 2018. We excluded 34% of the precipitation during the growing season, as given in Tab. 1, The scenarios for Switzerland (i.e., CH2018, https://www.meteoswiss.admin.ch/home/latest-news/news/climate-scenarios-ch2018.html; page 6) foresee a reduction up to 25% in precipitation in Switzerland in 2060 due to climate change, up to 40% by the end of the century. Our reduction for 2018 was right in the middle of these two numbers. In addition, the CH2018 scenarios foresee an increase of the longest rain-free summer period (June, July, August) from currently 11 days to 20 days, even shorter than our 5-week drought simulation. Thus, both length and severity of our treatment were very strong, clearly strong enough to simulate a future drought scenario threatening food security. We added further information in the discussion of the revised version. "The year 2018 was characterised by low precipitation during our experimental period, when a naturally dry period occurred at the end of our pronounced drought treatment in June (which excluded 34% of the precipitation during the growing season; Table 1). Our treatment compared well with the climate scenarios available for Switzerland, with a 25% reduction of precipitation in 2060, and up to 40%*

*by the end of the century; and an increase of the longest rain-free summer period (June, July, August) from currently 11 days to 20 days (CH2018)." (L387-392)*

- *(2) Regarding the natural drought: The naturally dry period 2018 occurred end of June, at the end of our experimental drought treatment (see Figs. 1 and 2), when the plants had already received about 150 mm of rain (see Table 1) and most likely also developed their main rooting system according to non-drought conditions. We added this into the revised manuscript, see our answer above. We expected differences among cropping systems because the Farming Systems and Tillage Experiment was established in 2009, i.e., the cropping systems were already in place 9 years before our drought study took place. We added this information to the MatMet (L128). It has been reported that these cropping systems in our site resulted in different soil properties (see Wittwer et al., 2021 which is now cited in the introduction; L50).*
  *Wittwer RA, Bender SF, Hartman K, Hydbom S, Lima RAA, Loaiza V, Nemecek T, Oehl F, Olsson PA, Petchey O, et al. 2021. Organic and conservation agriculture promote ecosystem multifunctionality. Science Advances 7(34): eabg6995.*

- *(3) Regarding the number of study years, we think this one year of experiment with an intensive sampling and measurement campaign is sufficient to show the response of a short-term crop like pea and barley and their water uptake to drought and the lack of cropping systems effects.*
  - *Our experimental design included drought and control sub-plots. Thus, we worked with replicated subplots in parallel (i.e., at the same time), not after each other (i.e., a temporal replication over multiple years). Since we were interested in the response of pea and barley to our drought treatment and not in their long-term yields, we think that our experimental design is adequate. We added this information in the MatMet Section. "Our experimental design thus compared replicated drought and control sub-plots in parallel (i.e., at the same time), not after each other (i.e., a temporal replication over multiple years), since in crop rotations, the identical crop cannot be grown on the same field for several years due to soil health issues." (L131-134)*
  - *In addition, in croplands, the identical crop should not be grown on the same field for several years due to arising soil health issues. This is the reason why agriculture in many countries works with rather elaborate crop rotation schemes, not only in Switzerland. Our study is no exception in this respect, a 6-yr rotation is used on this site and this rotation never has two legume crops following each other. So, it would not have been possible to again grow pea-barley in the following years on the same plots. We added further information to the MatMet (L131-134), see bullet above.*
  - *The increasing competition/lack of niche differentiation observed between pea and barley grown in a mixture under the drought treatment within the usual cropping season was surprising and deserves attention since farmers need to know about such an undesired outcome. This aspect is discussed in detail in L410-422.*
  - *Besides, the naturally dry period was not a disruption for the experiment. On the contrary, with this period (in control subplots, occurring at the end of the drought treatment) and the simulated drought (in drought subplots), we could study an aggravated drought scenario, one that is becoming more likely in the future (L113-116 and L387-392). See our answer above.*
- *(4) Regarding mycorrhizal fungi: We agree that mycorrhization is worth mentioning as one of the reasons that can cause different plant water uptake patterns in cropping systems. Some studies*

*discussed the positive effects of organic management and possibly from conservation tillage, maybe modulated via mycorrhizal fungi. Therefore, we now included this aspect (and reference: Wahdan et al., 2021) in the introduction (L45-46). Our results clearly showed no interactions between cropping systems and the simulated drought; and the potential benefits from soil as affected by different management practices were not able to overcome the drought effects (L423-439). Since we have no information on mycorrhization, and we consider mycorrhizal activities as part of those soil functions that were affected by our simulated drought, we did not expand the discussion on this particular soil function.*
*Wahdan SFM, Reitz T, Heintz-Buschart A, Schadler M, Roscher C, Breitkreuz C, Schnabel B, Purahong W, Buscot F. 2021. Organic agricultural practice enhances arbuscular mycorrhizal symbiosis in correspondence to soil warming and altered precipitation patterns. Environmental Microbiology 23(10): 6163-6176.*

Specific comments:

L13-17. Some of the Highlights bullets are unclear as written

- *Thank you for the comments. We guess that the lack of "was" or "were" created this impression. We rephrased the sentences (L15-16).*

L22-23. Which species is the important "fodder crop" or is it the mixture that is the "crop".

- *The 'pea-barley mixture' as an intercrop is the important fodder crop. We changed the sentence into "... a field-grown pea-barley (Pisum sativum L. and Hordeum vulgare L.) mixture, an important fodder intercrop, ..." (L23).*

L29-30. Check sentence construction.

- *We revised it as "Both species showed similar responses to the drought simulation and increased their proportional **water uptake** from shallow soil layer (0-20 cm) in all cropping systems." (L29-30).*

L42. "Adapted" should read "Adaptive"

- *Thank you for the suggestion. We changed it into "**Adaptive** crop management **to a changing climate**..." (L43)*

L63-70. Root-mycorrhizal interactions are also presumably important, but are not discussed here (and should be)

- *Thank you for the suggestion. We included mycorrhizal symbiosis as a potential aspect affected by different cropping systems (L45-46). See also response (4) above.*

L80-81. As opposed to when grown alone?

- *No, this is a misunderstanding. The aim was to compare the two species grown in mixture. We revised it as "if pea and barley differ in their water uptake patterns when grown in mixture, ..." (L81-82).*

L111-112. What are typical rainfall patterns during this period? Is a one-month drought during this time atypical? This seems like a fairly short period of drought conditions. Also, what was the

antecedent conditions in the months preceding the drought period. If rainfall was at or above average, perhaps a one month drought is not a problem? If I'm interpreting Table 1 correctly, it actually looks like precipitation in the control plots was ~30% of normal, so it appears there was a natural drought during the imposed drought period. I wonder why water wasn't added to control plots to simulate a "normal" year.

- *Thank you for the questions.*
  - o *Our study did not aim at an atypical or typical drought, but at a prolonged drought as predicted under climate change; the drought treatment was more than a month (37 d) and is therefore substantial for a three-and-a-half-month growing season for our pea-barley mixture. Currently, the longest period of no rain during summer in Switzerland is 11 days (see also our answer above). Our 37-day drought is therefore quite long. This is now clarified in MatMet: "In order to simulate a future drought scenario (CH2018), portable rain shelters were installed from 22 May to 28 June 2018 (37 days) during the 108-day growing season. This resulted in a 34% reduction in precipitation from the drought subplots during the growing season in 2018 (from sowing to harvest; **Error! Reference source not found.**)." (L113-116).*
  - o *Indeed, there was a naturally dry period, which was also addressed in the MatMet (L113-118; also see the point above) and the discussion (L387-391) and can be seen in Table 1 as well as in Figs 1 and 2. In Table 1, one can also see that May precipitation was similar to the long-term mean (102 vs 105 mm). So, no water surplus or scarcity occurred before our drought treatment. We added a sentence addressing this situation in the Results section to avoid any misunderstanding: "While the precipitation in May 2018 (102 mm) was comparable to the long-term mean (1988 to 2017: 105 mm), no precipitation fell between 14 June and 2 July 2018 (naturally dry period), resulting in a below-average precipitation in June (40 mm; long-term mean of 102 mm, Table 1), followed by an even more pronounced drought period in July (Fig. 1)." (L218-222).*
  - o *We did not irrigate the control subplot during the naturally dry period for different reasons: (i) We did not want to simulate a "normal" year vs. a drought period, but add the drought stress on top of the actual rainfed conditions. Additionally, irrigation is no typical agricultural measure in this region, particularly for such a crop. The naturally dry period was also not extraordinarily strong and happened in the ripening phase of the crop so that it would have caused substantial yield losses. (ii) We clearly did not anticipate such a naturally dry period during our experiment and adding irrigation short-term was beyond our logistic possibilities. We added a sentence in the MatMet section:"No irrigation was applied to the control plots during the (unexpected) naturally dry period in June for logistical and rational reasons, i.e., irrigation is unusual for the region and this crop, and the dry period happened during the ripening phase of the crop." (L116-118)*

L123-128. Describe antecedent moisture conditions here (i.e., per my questions above).

- *Thank you for the suggestion. However, we do not fully understand what is requested here in addition to what we already provide. In Table 1, we give the long-term climatic conditions as well as those during the growing season 2018 and separate months. In Figs 1 and 2 we show the temperature and the precipitation during the entire year 2018 (precipitation; Fig. 1) and during our experiment (soil moisture; Fig. 2). So, we think we have covered the antecedent moisture conditions already.*

L322-325. I don't understand this response. Under drought, when the surface soil is presumably dryer, wouldn't plants tend to have deeper rooting depths and/or obtain more water from depth?

- *Thank you for the question. Indeed, this question is what the Bayesian mixing models are used for, based on the stable water isotope data. While deeper rooting in response to drought is the usual assumption, experiments and observations do not always show such reliance on deeper soil layers during drought. We discussed such results in comparison to ours in detail in discussion (L349-364 and L365-386).*

L415. "adaptive" not "adapted"

- *Thank you for the correction. We changed the word accordingly (L448).*

L415-417. I would take some care with this conclusion, acknowleding that cropping system differences may be more pronounced under longer-term drought (see previous comments above)

- *Thank you for the suggestion. As you have pointed out above, in addition to our drought treatment, there was a naturally dry period (leading to only 40% of monthly precipitation in June 2018 compared to long-term average in June under rainfed conditions). Our 37-d drought treatment simulated a much more severe drought than the naturally dry period (excluding 79 mm). Altogether, we report data of a "more pronounced drought". Still, we did not find significant differences in changing water uptake patterns among the different cropping systems. We consider our results sufficient to conclude that under future climate with similarly small water supply, cropping systems might not be reliable to provide consistent alleviation for plant drought stress for pea-barley. But as written in L439-442 and L450-452, it remains to be seen how other crops react.*

Table 1. It unclear what the last three lines in the table refer to. Are these precipitation amounts in the drought treatment itself?

- *As indicated in the first column, the values are total precipitation and mean air temperature during the respective time periods separated by the drought treatment. Hence, not just in the drought treatment itself, but before, during, and after the drought treatment. We revised the second last cell to make this clearer (changed "End of drought treatment" to "During drought treatment").*

Table 2-5. I assume the values shown are P-values, but that is not explicilty stated in the table heading. This is a lot of tables of p values...

- *Thank you for the comment. Yes, they are p values. We revised the captions to clarify the values.*

L671-674. I'm unclear why these lines are shown below the table. Is this text part of the heading or a footnote to the table?

- *It serves as a footnote to the table.*